# IEDR: A Context-aware Intrinsic and Extrinsic Disentangled Recommender System

## Abstract

Intrinsic and extrinsic factors jointly affect users' decisions in item selection (e.g., click, purchase). Intrinsic factors reveal users' real interests and are invariant in different contexts (e.g., time, weather), whereas extrinsic factors can change w.r.t. different contexts. Analyzing these two factors is an essential yet challenging task in recommender systems. However, in existing studies, factor analysis is either largely neglected, or designed for a specific context (e.g., the time context in sequential recommendation), which limits the applicability of such models. In this paper, we propose a generic model, IEDR, to learn intrinsic and extrinsic factors from various contexts for recommendation. IEDR contains two key components: a contrastive learning component, and a disentangling component. The two components collaboratively enable our model to learn context-invariant intrinsic factors and context-based extrinsic factors from all available contexts. Experimental results on real-world datasets demonstrate the effectiveness of our model in factor learning and impart a significant improvement in recommendation accuracy over the state-of-the-art methods.

## 1 Introduction

Recommender systems aim to predict the probability of a user selecting a given item (e.g., click, purchase). This is a challenging prediction as each decision is jointly affected by multiple factors (Ma et al., 2019). Psychological research has revealed that users' decision making is mainly influenced by two factors: *intrinsic* and *extrinsic* factors (Bénabou & Tirole, 2003; Vallerand, 1997). An intrinsic factor is an internal motivation for inherent satisfaction, which is often stable for an individual. In contrast, an extrinsic factor is a contextual motivation triggered by the environment (external stimulation), and it often varies among different contexts (e.g., weather, time) (Ryan & Deci, 2000). For example, on a day with heavy rain, a user decides to take an Uber (a taxi calling app) to work. In this case, the choice of Uber over other taxi calling apps is because the user is more comfortable with this app's user interface (intrinsic factor), while the choice of taking a ride to work is motivated by the weather condition (extrinsic factor).

Although the importance of capturing these factors in recommender systems has been recognized, their full potential has not been explored by the existing works. (1) Some studies neglect the intrinsic and extrinsic factor disentangling, and the final prediction mainly relies on learning entangled representations (Barkan & Koenigstein, 2016; Covington et al., 2016; Wu et al., 2019). With the intrinsic and extrinsic factors entangled behind each decision, the real factors that derive the decision may be incorrectly inferred, resulting in a suboptimal recommendation Wang et al. (2020). (2) Some studies learn intrinsic and extrinsic factors, but just under a specific context. For example, some sequential recommendation models leverage the time context (order sequence) to learn intrinsic and extrinsic factors (they call them long- and short-term interests) (Hidasi et al., 2016; Yu et al., 2019b); some point-of-interest recommendation models leverage the spatial context (geometric distance) to learn the two factors (Li et al., 2017; Wu et al., 2020). In such models, the factor learning approaches are domain-specific, so it would be difficult to generalize them to other contexts. Meanwhile, the factors may be influenced by multiple contexts. Hence, focusing on a single context may result in inferior factor learning. Therefore, it is still an open question of how to effectively incorporate various context information for learning intrinsic and extrinsic factors in recommender systems.

Focusing on this question, we propose a generic recommendation framework that can learn intrinsic and extrinsic factors from various contexts. We first formally define context-agnostic intrinsic and extrinsic factors for recommendation tasks. Following these definitions, we propose an Intrinsic-Extrinsic Disentangled Recommendation (IEDR) model, which contains two modules: a recommendation prediction (RP) module, and a contrastive intrinsic-extrinsic disentangling (CIED) module. For each user-item interaction, the PR module constructs all the contexts as a graph (context graph), and the context representation is obtained via learning the graph. The same procedures are done to obtain the user and item representations from their attributes (e.g., user gender, item category), respectively. Then, the intrinsic and extrinsic factors are learned from these representations for the user and the item perspectives. Meanwhile, the CIED module contains two components: a contrastive learning component that learns a context-invariant intrinsic factor, and a disentangling component that disentangles the intrinsic and extrinsic factors via mutual information minimization. The two components jointly ensure IEDR to learn intrinsic and extrinsic factors.

In this paper, we made the following contributions:

- To better analyze the factors influencing users' decisions, we formalize the context-agnostic intrinsic and extrinsic factors for recommender systems. Following these definitions, we propose IEDR to learn disentangled intrinsic and extrinsic factors from various contexts for recommendation.

- IEDR comprises a context-invariant contrastive learning component, and a mutual information minimization-based disentangling component to effectively disentangle the learned factors.

- Extensive experiments on real-world datasets show that (1) IEDR significantly outperforms the state-of-the-art baselines when various contexts are available; (2) our proposed CIED module can successfully learn intrinsic and extrinsic factors.

## 2 RELATED WORK

This section summarizes the current research progress on recommender systems and contrastive learning related to our work.

**Feature interaction modeling**   Many recommender systems leverage feature interactions to improve prediction accuracy. One of the most common techniques is the factorization machine (FM) (Rendle, 2010), which models feature interactions through dot product and achieves great success. Recent studies extend FM with deep neural networks for more powerful feature interaction modeling (Xiao et al., 2017; He & Chua, 2017; Yu et al., 2019a). The Wide & Deep model (WDL) (Cheng et al., 2016) proposes a framework that combines shallow and deep modeling of features for recommendation. Guo et al. (2017) combine FM and WDL by replacing the shallow part of WDL with an FM model. Su et al. (2021) leverage the relation reasoning power of graph neural networks for feature interaction modeling. However, these models do not incorporate context information for better factor analysis, and we overcome this issue by leveraging this information to disentangle and learn intrinsic and extrinsic factors for recommendation.

**Factor disentanglement**   Intrinsic and extrinsic factors are considered as two basic factors for individual decision making in psychological research (Ryan & Deci, 2000; Bénabou & Tirole, 2003; Vallerand, 1997). Recent recommender systems have borrowed the idea of capturing these two factors to achieve more accurate recommendation. For example, in sequential recommendation, Hidasi et al. (2016) are the first to leverage the recurrent neural networks to capture users' long- and short-term (LS-term) interests from their interacted item sequences. Yu et al. (2019b) propose a time-aware controller to capture the differences between LS-term interests for more accurate interest learning. Zheng et al. (2022) further emphasize the disentanglement between the LS-term interests at different time scales to differentiate the LS-term interests. In point-of-interest recommendation, studies are leveraging spatial context to capture the intrinsic and extrinsic factors (Li et al., 2017; Wu et al., 2020). However, all of the above studies focus on specific contexts. As a result, their factor learning approaches are hard to apply to other recommendation domains, which may result in a suboptimal solution if other contexts jointly influence these factors. Some studies learn users' multiple factors without knowing the meaning of each factor (i.e., implicit factor). They first define the number of factors (e.g., 4) to be learned, and then disentangle the representations of each pair of factors (Ma et al., 2019; Wang et al., 2020). These models only ensure that the learned factor

representations are disentangled, but cannot guarantee whether they really refer to important factors. Our IEDR model incorporates various contexts for explicit intrinsic and extrinsic factor learning.

**Contrastive learning**   Contrastive learning has achieved great success in computer vision (Chen et al., 2020; Chuang et al., 2020; Khosla et al., 2020; Tian et al., 2020; Chen & He, 2021) and neural language processing (Oord et al., 2018; Yang et al., 2019; Gao et al., 2021; Gunel et al., 2021). Recently, contrastive learning has attracted attention in recommender systems. Yao et al. (2021) conduct contrastive learning on users and items respectively on a two-tower framework to learn robust user and item representations. In addition, Wu et al. (2021) propose a contrastive learning framework on a user-item bipartite graph to capture robust high-degree relationships between users and items. Lin et al. (2021) and Jiang et al. (2021) leverage contrastive learning to eliminate popularity bias. We propose the method that learns intrinsic factor representations that are invariant to context through a contrastive learning approach.

## 3   PROBLEM STATEMENT AND DEFINITIONS

Let $\mathcal{U}$, $\mathcal{V}$, and $\mathcal{C}$ denote the user set, item set, and context set, respectively. Each user $u \in \mathcal{U}$ consists a set of user features $u = \{z_1^u, z_2^u, ..., z_p^u\}$ (e.g., user ID, gender). Similarly, each item $v \in \mathcal{V}$ is represented by a set of item features $v = \{z_1^v, z_2^v, ..., z_q^v\}$ (e.g., branch, color). A context $c \in \mathcal{C}$ is a set of context features $c = \{z_1^c, z_2^c, ..., z_m^c\}$, denoting the context state when a user selects an item (e.g., weather, daytime). Let $\mathcal{D}$ be a dataset containing $N$ instances (i.e., data samples) of $(u, v, c)$, with an corresponding label $y \in \{1, 0\}$ indicating whether or not the user $u$ selects the item $v$ under the context $c$. The recommendation task can be formulated as predicting the selection probability $y' = p(u, v, c)$. In our proposed IEDR model, the intrinsic factor $\boldsymbol{o}_{in}$ and the extrinsic factor $\boldsymbol{o}_{ex}$ are explicitly inferred for both users and items, and jointly leveraged to perform the prediction.

Next, we formally define intrinsic and extrinsic factors. We believe these two factors exist from both users' and items' perspectives. This is reasonable since a user selecting an item not only relates to the factors (motivations) of users, e.g., *preferring a ride* (intrinsic factor) over walking to work on a *rainy day* (extrinsic factor), but also relates to the factors (attractiveness) of items, e.g., a Taxi calling App with a *comfortable user interface* (intrinsic factor) and *has a discount* (extrinsic factor). In the following, we define intrinsic and extrinsic factors from the users' perspective only, as they are similar from the items' perspective.

**Definition 1.** *(**Intrinsic Factor and Extrinsic Factor**) Consider a user $u$ and a set of contexts $\mathcal{C}$; an **intrinsic factor** of the user $u$ is a factor that is invariant to the contexts in $\mathcal{C}$, i.e., $f_{in}(u, c) = f_{in}(u, c')$, where $f_{in}$ is a function learning intrinsic factor representations, and $c$ and $c'$ are two arbitrary contexts in $\mathcal{C}$. On the other hand, an **extrinsic factor** of the user $u$ is a factor that changes w.r.t. the context, i.e., there exist contexts $c$ and $c'$ in $\mathcal{C}$ such that $f_{ex}(u, c) \neq f_{ex}(u, c')$, where $f_{ex}$ learns extrinsic factor representations.*

## 4   INTRINSIC-EXTRINSIC DISENTANGLED RECOMMENDATION MODEL

The overview of our model is visualized in Figure 1. More specifically, our proposed IEDR model consists of the following two modules, which will be detailed in the next subsections:

- A recommendation prediction (RP) module that takes a user and an item as input, and combines them with a set of contexts, to generate intrinsic and extrinsic factor representations for both the user and the item. The predicted probability $y'$ is then jointly learned from these representations.
- A contrastive intrinsic-extrinsic disentangled (CIED) module is applied to both the user and the item sides to support the intrinsic and extrinsic factor learning. The module contains a context-invariant contrastive learning component and a disentangling component, to ensure the learned factors satisfy Definition 1.

### 4.1   THE RECOMMENDATION PREDICTION MODULE

The recommendation prediction (RP) module is a symmetric structure that generates user intrinsic and extrinsic factor representations $(\boldsymbol{o}_{in}^u, \boldsymbol{o}_{ex}^u)$ from the user side, and generates item intrinsic and

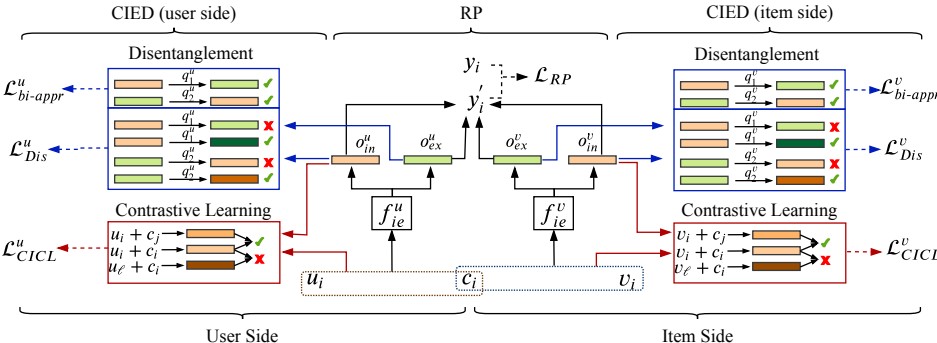

Figure 1: An Overview of IEDR. It is a symmetric structure on the user side and the item side. The middle part (the black arrows) represents the recommendation prediction (RP) module (Section 4.1). It generates the intrinsic and extrinsic factor representations ($\boldsymbol{o}_{in}$ and $\boldsymbol{o}_{ex}$) for producing the recommendation prediction $y'$. The side parts are two contrastive intrinsic-extrinsic disentanglement (CIED) modules. Each CIED includes a context-invariant contrastive learning component (the red arrows, Section 4.2.1), and a disentangling component (the blue arrows, Section 4.2.2) to ensure the success of the factor learning. The losses generated through these modules ($\mathcal{L}_{RP}, \mathcal{L}_{CICL}, \mathcal{L}_{bi-appr}, \mathcal{L}_{Dis}$) will be optimized as a two-step multi-task training (Section 4.3).

extrinsic factor representations ($\boldsymbol{o}_{in}^v, \boldsymbol{o}_{ex}^v$) from the item side. On the user side, we first generate a user representation and a context representation based on user features and context features, respectively. Here we use the SIGN model (Su et al., 2021) to generate the representations. SIGN has been proven effective in user/item/context representation learning through modeling feature interactions via graph neural networks. Appendix A.1 provides a detailed description of SIGN. More formally, let $f_u(u) : \mathbb{R}^{p \times d} \to \mathbb{R}^d$ be the function for SIGN-based feature modeling. $f_u(u)$ first maps each user feature $z_i^u \in u$ into a $d$-dimensional feature embedding $\boldsymbol{z}_i^u$. Then, it models these feature embeddings to output the user representation $\boldsymbol{u}$. Similarly, SIGN learns context representation $\boldsymbol{c}$ through $f_c$. Next, a factor generation function $f_{ie}^u(\boldsymbol{u}, \boldsymbol{c}) : \mathbb{R}^{2 \times d} \to \mathbb{R}^{2 \times d}$ (e.g., a neural network) takes the user representation and the context representation as input, and simultaneously generates a user intrinsic representation $\boldsymbol{o}_{in}^u$ and a user extrinsic representations $\boldsymbol{o}_{ex}^u$. Here, the output is a $2d$-dimensional vector, with the first $d$-dimensional terms as $\boldsymbol{o}_{in}^u$ and the rest as $\boldsymbol{o}_{ex}^u$. On the item side, a similar module structure is adopted. We use a different SIGN-based function for the item representation learning $\boldsymbol{v} = f_v(v)$, while using the same context representation as that on the user side. A factor generating function $f_{ie}^v(\boldsymbol{v}, \boldsymbol{c})$ is applied to obtain the item intrinsic factor representation $\boldsymbol{o}_{in}^v$ and extrinsic factor representation $\boldsymbol{o}_{ex}^v$.

Finally, we jointly learn the prediction $y' = f_{pred}(\boldsymbol{o}_{in}^u, \boldsymbol{o}_{ex}^u, \boldsymbol{o}_{in}^v, \boldsymbol{o}_{ex}^v)$. We linearly combine the intrinsic and extrinsic representations and use the dot product as the prediction function: $f_{pred}(\boldsymbol{o}_{in}^u, \boldsymbol{o}_{ex}^u, \boldsymbol{o}_{in}^v, \boldsymbol{o}_{ex}^v) = (\boldsymbol{o}_{in}^u + \boldsymbol{o}_{ex}^u)^\top (\boldsymbol{o}_{in}^v + \boldsymbol{o}_{ex}^v)$. A cross-entropy loss function is adopted to minimize the prediction error: $\mathcal{L}_{RP}(u, v, c) := -y \log(y') + (1 - y) \log(1 - y')$.

## 4.2 THE CONTRASTIVE INTRINSIC-EXTRINSIC DISENTANGLING MODULE

While the RP module can generate factor representations, solely using this module cannot correctly distinguish intrinsic representations from extrinsic ones. To address this, we propose a contrastive intrinsic-extrinsic disentangled (CIED) module and apply it to both the user and the item sides. In the following, we only describe the CIED on the user side, as the module on the item side has the same structure. CIED consists of a *context-invariant contrastive learning component* and a *disentangling component*. The contrastive learning component learns intrinsic representations that are invariant in different contexts, while the disentangling component leverages a mutual information minimization task to disentangle the intrinsic and extrinsic representations. In the following, we describe these two components in detail.

### 4.2.1 THE CONTEXT-INVARIANT CONTRASTIVE LEARNING COMPONENT

We propose a context-invariant contrastive learning component to learn the intrinsic representations. More specifically, we maximize the agreement between the intrinsic representation pairs generated from the same user under different contexts (positive pairs), and minimize the agreement between the

intrinsic representation pairs generated from the same context with different users (negative pairs) at the same time. More formally, we represent the intrinsic representations with the subscript $(o_{in}^u)_{ij}$ if it is generated through user $u_i$ (from $i$-th data sample) and context $c_j$ (from $j$-th data sample), i.e., $(o_{in}^u)_{ij} = f_{ie}^u(u_i, c_j)$. For the $i$-th data sample $(u_i, v_i, c_i) \in \mathcal{D}$, we calculate the objective function based on InfoNCE (Oord et al., 2018):

$$\mathcal{L}_{\text{CICL}}^u(u_i, c_i) := -\log \frac{\exp\left(\text{sim}((o_{in}^u)_{ii}, (o_{in}^u)_{ij})/\tau\right)}{\sum_{u_\ell \in \mathcal{U}} \exp\left(\text{sim}((o_{in}^u)_{ii}, (o_{in}^u)_{\ell i})/\tau\right)}, \tag{1}$$

where $c_j$ is an arbitrary context, $\text{sim}(\cdot)$ is the cosine similarity, and $\tau$ is a temperature value.

**Implementation**. To optimize Equation (1), we need to generate $c_j$ ($c_i \neq c_j$). We adopt a simple method to perform random sampling from the contexts within the same batch. Also, inspired by (Gao et al., 2021), where a vector with a large dropout rate can be considered as a new vector, we propose a dropout-based method to generate new context representations (e.g., with the dropout rate larger than 50%). In practice, we integrate the two methods to generate different contexts $c_j$ for each $c_i$ (we empirically show in Appendix G.4 that the integrated generating method results in better prediction accuracies than either methods). Meanwhile, we need to iterate over all the users to generate $(o_{in}^u)_{\ell i}$ (the intrinsic representations generated from an arbitrary user $u_\ell$ and context $c_i$), which is prohibitive when the users' number is large. Here, we randomly sample $L$ users from the same batch to generate the negative intrinsic representations $(o_{in}^u)_{\ell i}$, where $\ell = 1, 2, \cdots, L$. We use the categorical cross-entropy to optimize Equation (1) following (Oord et al., 2018).

### 4.2.2 THE DISENTANGLING COMPONENT

We then perform an intrinsic-extrinsic factor disentangling via minimizing the mutual information between the $o_{in}^u$ and $o_{ex}^u$ generated from $f_{ie}^u(u, c)$. Inspired by vCLUB (Cheng et al., 2020), we minimize their mutual information by estimating a vCLUB-based upper bound. However, the asymmetric property of the original vCLUB may lead to a less robust and inferior intrinsic-extrinsic disentangling (further discussions on this drawback of vCLUB can be found in Appendix D). Therefore, we propose a simple yet effective bidirectional extension of vCLUB for symmetry, which is more robust and achieves better disentanglement. In the bidirectional vCLUB, two variational distributions (e.g., approximated via neural networks) $q_1^u(o_{ex}^u|o_{in}^u; \theta_1^u)$ and $q_2^u(o_{in}^u|o_{ex}^u; \theta_2^u)$ are proposed with parameters $\theta_1^u$ and $\theta_2^u$, to predict the two types of factors, respectively. Then a bidirectional vCLUB-based mutual information upper bound can be obtained as:[1]

$$\begin{aligned}
\mathcal{I}_{\text{bi-vCLUB}}(o_{in}^u; o_{ex}^u) := \frac{1}{2}\Big( & \mathbb{E}_{p(o_{in}^u, o_{ex}^u)}[\log q_1^u(o_{ex}^u|o_{in}^u)] - \mathbb{E}_{p(o_{in}^u)p(o_{ex}^u)}[\log q_1^u(o_{ex}^u|o_{in}^u)] \\
& + \mathbb{E}_{p(o_{in}^u, o_{ex}^u)}[\log q_2^u(o_{in}^u|o_{ex}^u)] - \mathbb{E}_{p(o_{ex}^u)p(o_{in}^u)}[\log q_2^u(o_{in}^u|o_{ex}^u)]\Big).
\end{aligned} \tag{2}$$

Through minimizing the bidirectional upper bound $\mathcal{I}_{\text{bi-vCLUB}}(o_{in}^u; o_{ex}^u)$ as above, we minimize the mutual information between $o_{in}^u$ and $o_{ex}^u$. Experimental results in Section 5.2.2 show that vCLUB is more robust and achieves better factor learning.

**Implementation**. The optimization of the disentangling component is conducted in two steps iteratively. In the first step, we estimate the upper bound by training $\theta_1^u$ and $\theta_2^u$ to minimize the loss function $\mathcal{L}_{bi-appr}^u(u_i, c_i) := -\frac{1}{2}\Big(\log q_1^u\big((o_{ex}^u)_{ii}|(o_{in}^u)_{ii}\big) + \log q_2^u\big((o_{in}^u)_{ii}|(o_{ex}^u)_{ii}\big)\Big)$. Following (Cheng et al., 2020), we use the mean squared error to optimize $q_1^u$ and $q_2^u$. In the second step, we freeze $\theta_1^u$ and $\theta_2^u$, and minimize the mutual information of $o_{in}^u$ and $o_{ex}^u$ by training other parameters to minimize the upper bound $\mathcal{L}_{Dis}^u(u_i, c_i) = \mathcal{I}_{\text{bi-vCLUB}}\big((o_{in}^u)_{ii}; (o_{ex}^u)_{ii}\big)$.

### 4.3 A MULTI-TASK TRAINING

We perform a two-step multi-task training to minimize the empirical risk of IEDR. The two steps run alternatively until convergence. Appendix H provides the pseudo-code of the training procedure. In the first step, we freeze all the parameters except for $\theta_1^u, \theta_2^u, \theta_1^v$, and $\theta_2^v$, where $\theta_1^v, \theta_2^v$ are the parameters of $q_1^v(o_{ex}^v|o_{in}^v; \theta_1^v)$ and $q_2^v(o_{in}^v|o_{ex}^v; \theta_2^v)$ in the disentangling component on the item side.

---

[1] $\mathcal{I}_{\text{bi-vCLUB}}(o_{in}^u; o_{ex}^u)$ is the average of two vCLUB-based upper bounds of different directions. Therefore, it is obvious that $\mathcal{I}_{\text{bi-vCLUB}}(o_{in}^u; o_{ex}^u)$ is still an upper bound of $\mathcal{I}(o_{in}^u; o_{ex}^u)$.

We then minimize $\mathcal{R}(\boldsymbol{\theta}_1^u, \boldsymbol{\theta}_2^u, \boldsymbol{\theta}_1^v, \boldsymbol{\theta}_2^v) = \frac{1}{N}\sum_{i=1}^N \left(\mathcal{L}_{bi\text{-}appr}^u(u_i, c_i) + \mathcal{L}_{bi\text{-}appr}^v(v_i, c_i)\right)$. In the second step, we freeze $\boldsymbol{\theta}_1^u, \boldsymbol{\theta}_2^u, \boldsymbol{\theta}_1^v$, and $\boldsymbol{\theta}_2^v$, and minimize the following function:

$$\arg\min \mathcal{R}(\boldsymbol{\omega}) = \frac{1}{N}\sum_{i=1}^N \Big(\mathcal{L}_{\text{RP}}(u_i, v_i, c_i) + \lambda_1\big(\mathcal{L}_{\text{CICL}}^u(c_i, u_i) + \mathcal{L}_{\text{CICL}}^v(c_i, v_i)\big) + \lambda_2\big(\mathcal{L}_{Dis}^u(u_i, c_i) + \mathcal{L}_{Dis}^v(v_i, c_i)\big)\Big), \quad (3)$$

where $\mathcal{L}_{bi\text{-}appr}^v$, $\mathcal{L}_{\text{CICL}}^v$, and $\mathcal{L}_{Dis}^v$ are the losses on the item side, $\lambda_1$ and $\lambda_2$ are the weight factors, and $\boldsymbol{\omega}$ are all the trainable parameters except for $\boldsymbol{\theta}_1^u, \boldsymbol{\theta}_2^u, \boldsymbol{\theta}_1^v$, and $\boldsymbol{\theta}_2^v$.

### 4.4 THEORETICAL ANALYSIS: CONTEXT-INVARIANT CONTRASTIVE LEARNING IN INFORMATION THEORY

In this section, we reason the context-invariant contrastive learning from the perspective of information theory. As formally defined in Theorem 1, optimizing Equation (1) is equivalent to maximizing the mutual information between the intrinsic representations and user representations, and simultaneously minimizing the mutual information between the intrinsic representations and the context representations. The theorem on the item side can be derived in the same fashion. The proof of this equivalence can be found in Appendix B.

**Theorem 1** (Equivalence of contrastive loss $\mathcal{L}_{CICL}^u$). *Optimizing the contrastive loss is equivalent to solving:*

$$\arg\min \sum_{i=1}^N \mathcal{L}_{CICL}^u(u_i, c_i) = \arg\max\Big(\mathcal{I}(\boldsymbol{o}_{in}^u, \boldsymbol{u}) - \mathcal{I}(\boldsymbol{o}_{in}^u, \boldsymbol{c})\Big). \quad (4)$$

## 5 EXPERIMENTS

We conduct extensive experiments to demonstrate the effectiveness of our model. In this section, we focus on 1) the recommendation performance of IEDR compared to the state-of-the-art methods; 2) the effectiveness of each component in IEDR; and 3) the ability to disentangle intrinsic and extrinsic factors of IEDR. We discuss the datasets, baselines, implementations, and parameter settings in detail in Appendix F.

### 5.1 OVERALL PERFORMANCE

We evaluate the recommendation performance of our model, by comparing it with various baselines in two scenarios. In the first scenario, we learn intrinsic and extrinsic factors from various contexts. In the second scenario, we learn the factors from a specific (time) context and compare with sequential recommendation baselines that also learn intrinsic and extrinsic factors (i.e., long and short term interests). We use three common evaluation metrics for recommender systems: AUC, NDCG@$k$, and HR@$k$ with $k$ being 5 and 10.

### 5.1.1 FACTOR LEARNING FROM VARIOUS CONTEXTS

We run our model and feature interaction-based baselines on the datasets that contain various contexts: the Frappe dataset (Baltrunas et al., 2015) for app recommendation, and the Yelp dataset (Wu et al., 2022) for restaurant recommendation. As our baselines, we use feature interaction-based approaches (Xiao et al., 2017; He & Chua, 2017; Song et al., 2019; Guo et al., 2017; Cheng et al., 2016; Wang et al., 2021; Yu et al., 2019a; Su et al., 2021) that capture these contexts but neglect the factor learning. We also compare with implicit factor disentanglement methods DisRec (Ma et al., 2019) and DGCF (Wang et al., 2020). The performance results are reported in Table 1. Each result is the average of 10 times run. For each metric, the best results are in bold, and the best baseline results are underlined. The rows *Improv* (standing for Improvements) and *p-value* show the improvement and statistical significance (via the Wilcoxon signed-rank test (Wilcoxon, 1992)) of IEDR over the best baseline results, respectively.

From Table 1, we observe that our model significantly outperforms all baselines under all evaluation metrics, and the *p-value*s are all less than the 5% threshold, indicating the significance of the improvements. IEDR outperforms feature interaction-based baselines because IEDR captures the

Table 1: Comparing the prediction performance (in percentage) with the baselines. The *Improv* and *p-value* rows show the relative improvements and the statistical significance of IEDR over the best performed baselines, respectively. N@k refers to NDCG@k and H@k refers to HR@k.

| | Frappe | | | | | Yelp | | | | |
|---|---|---|---|---|---|---|---|---|---|---|
| | AUC | N@5 | N@10 | H@5 | H@10 | AUC | N@5 | N@10 | H@5 | H@10 |
| AFM | 93.18 | 63.52 | 67.44 | 77.84 | 84.71 | 91.96 | 42.79 | 47.17 | 58.69 | 72.21 |
| NFM | 95.86 | 68.30 | 70.73 | 83.00 | 90.40 | 93.32 | 45.99 | 50.33 | 61.90 | 75.27 |
| AutoInt | 95.83 | 69.45 | 71.41 | 84.04 | 90.10 | 93.82 | 46.61 | 50.80 | 63.72 | 76.55 |
| DeepFM | 96.09 | 69.20 | 71.28 | 82.70 | 89.50 | 93.26 | 44.20 | 48.50 | 60.26 | 73.55 |
| WDL | 95.96 | 68.02 | 70.33 | 81.70 | 88.90 | 93.41 | 45.47 | 49.71 | 61.90 | 74.89 |
| DCNv2 | 95.25 | 68.15 | 70.34 | 82.15 | 89.91 | 93.66 | 43.41 | 48.26 | 60.97 | 74.88 |
| IFM | 95.32 | 66.91 | 69.13 | 80.90 | 87.60 | 93.83 | 46.74 | 50.86 | 63.04 | 75.69 |
| SIGN | 95.92 | 69.38 | 71.49 | 83.91 | 90.37 | 93.67 | 46.80 | 50.94 | 63.68 | 76.41 |
| DisRec | 85.51 | 56.81 | 60.07 | 67.42 | 76.29 | 84.01 | 34.82 | 37.90 | 48.29 | 63.17 |
| DGCF | 86.13 | 58.40 | 61.44 | 69.05 | 77.53 | 85.29 | 36.35 | 39.06 | 50.05 | 64.62 |
| IEDR | **96.34** | **72.40** | **74.11** | **85.94** | **91.25** | **94.22** | **48.68** | **53.05** | **65.23** | **78.29** |
| *Improv* | 0.26% | 4.24% | 3.66% | 2.26% | 0.94% | 0.42% | 4.01% | 4.14% | 2.38% | 2.28% |
| *p-value* | 3.72% | 0.25% | 0.25% | 0.25% | 0.83% | 2.34% | 0.25% | 0.25% | 0.25% | 0.25% |

intrinsic and extrinsic factors, while these baselines neglect the factor learning. Meanwhile, the implicit factor disentanglement methods (DisRec and DGCF) also perform inferior to our model. One reason is that implicit factor disentanglement is not the best way to infer these factors. In Section 5.2, we empirically verify that replacing the disentanglement module in IEDR with an implicit approach (as in DisRec and DGCF) leads to a decrease in recommendation accuracy. Another reason is that the factor disentanglement of DisRec and DGCF are purely based on user-item interactions, and do not consider context information. This may lead to critical information loss for recommendation. Our model explores context information to learn disentangled intrinsic and extrinsic factors for recommendation, and hence achieves better prediction accuracy.

### 5.1.2 FACTOR LEARNING FROM SPECIFIC CONTEXT

We then evaluate IEDR on two Amazon datasets (Movies and CDs) (McAuley et al., 2015) that contain only the time context. IEDR takes the (bucketed) time context as features to learn intrinsic and extrinsic factors. We compare with the state-of-the-art sequential recommendation baselines (Hidasi et al., 2016; Yu et al., 2019b; Zheng et al., 2022) that learn LS-term interests from the item sequences ordered by the time. The experimental results are reported in Table 2, where our model achieves competitive accuracy compared to the baselines. This proves the ability of our model to achieve state-of-the-art recommendation accuracy in the context-specific scenario, even compared with the models designed for the context. Moreover, our IEDR is more versatile and can be applied to various contexts.

Table 2: Comparing the performance of IEDR and the baselines on time context-specific scenario.

| | Movies | | CDs | |
|---|---|---|---|---|
| | AUC | N@10 | AUC | N@10 |
| GRU4Rec | 77.11 | 25.18 | 78.86 | 19.41 |
| SLI-Rec | 78.69 | 26.85 | 79.37 | 20.27 |
| CLSR | 80.02 | 26.98 | 80.42 | 21.07 |
| IEDR | 80.14 | 26.68 | 80.34 | 20.95 |

### 5.2 EFFECTIVENESS OF OUR MODEL'S COMPONENTS

This section evaluates the components in IEDR in detail. We demonstrate the results in NDCG@10 due to the space limit. Other metrics show similar trends.

### 5.2.1 ABLATION STUDY OF CONTRASTIVE INTRINSIC-EXTRINSIC DISENTANGLING MODULE

The contrastive intrinsic-extrinsic disentangling (CIED) module contains a context-invariant contrastive learning component and a disentangling component. In this section, we conduct an ablation study to show the impact of these components. We run our model in three variations: 1) without the

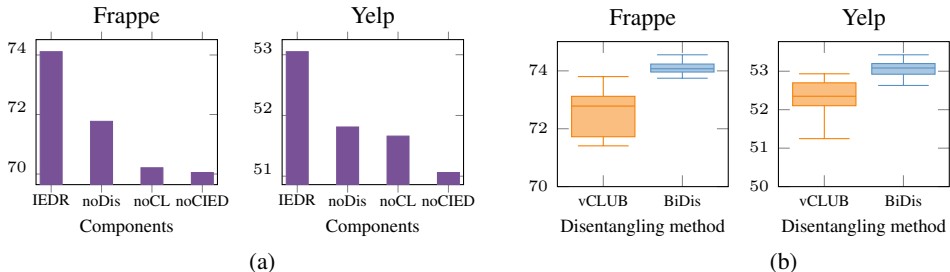

Figure 2: (**a**) Ablation studies with different component(s) removed. (**b**) The performance and variance statistics of vCLUB and BiDis. The vertical axis is NDCG@10.

contrastive learning component (*noCL*); 2) without the disentangling component (*noDis*); 3) without the contrastive learning and disentangling components (*noCIED*), i.e., the CIED module is not applied. Figure 2a compares our IEDR model with these three variants. The inferior performance of *noCIED* compared with our full model *IEDR* demonstrates the importance of learning intrinsic and extrinsic factors for accurate recommendation prediction. *noCL* can be regarded as performing implicit factor learning. The inferior performance of *noCL* indicates that explicit intrinsic and extrinsic factor learning is superior to implicit factor learning methods (as in the factor disentangling baselines) in inferring the real reason for users' decisions. *noDis* learns intrinsic factors but does not guarantee the extrinsic factor learning. Therefore, it also obtains worse results than CIED. Employing both components achieves better performance than learning with only one. This is because either component cannot individually learn intrinsic and extrinsic factors successfully, highlighting the importance of learning the two factors for an accurate recommendation.

### 5.2.2 DISENTANGLING COMPONENT EVALUATION

We propose a bidirectional vCLUB-based disentangling method (*BiDis*) that disentangles the intrinsic and extrinsic factors. In this section, we compare our *BiDis* method with the original vCLUB method (*vCLUB*). From Figure 2b, we observe that our *BiDis* achieves a better performance than *vCLUB*, and its variance is much smaller than *vCLUB*. This indicates that *BiDis* generates more robust predictions, which is consistent with our analysis in Section 4.2.2. In addition, we visualize the intrinsic and extrinsic representations learned by the two disentanglement methods with t-SNE in Appendix G.7, and observe that using *BiDis* results in a better intrinsic-extrinsic disentanglement. This is the reason why *BiDis* delivers a better performance than *vCLUB*.

### 5.3 DISENTANGLEMENT VERIFICATION

This section verifies the intrinsic and extrinsic factor disentangling ability of IEDR, including a visualization of the learned intrinsic and extrinsic representations and a case study to show the differences of these factors in users' decision-making.

### 5.3.1 VISUALIZATION OF INTRINSIC AND EXTRINSIC REPRESENTATIONS

In Figure 3, we visualize intrinsic and extrinsic factor representations learned from our model. We see that when the model is equipped with the CIED module (*IEDR*), the factors are well disentangled. However, when we do not use the CIED module (*noCIED*), the intrinsic and extrinsic factor representations are mixed together. This indicates that these factors cannot be well learned and disentangled without our CIED module. We provide more visualizations and analysis for our model with different component combinations in Appendix G.7.

### 5.3.2 CASE STUDY

We conducted a case study to analyze the differences between the learned intrinsic and extrinsic factors. We randomly choose a user from the Frappe dataset and generate the intrinsic matching scores (the dot product of the user's intrinsic representation and the items' (apps) intrinsic representations) in two different contexts (Weekday and Weekend). The same for the extrinsic matching scores. We sort the matching scores for the intrinsic and extrinsic factors, respectively, and list the

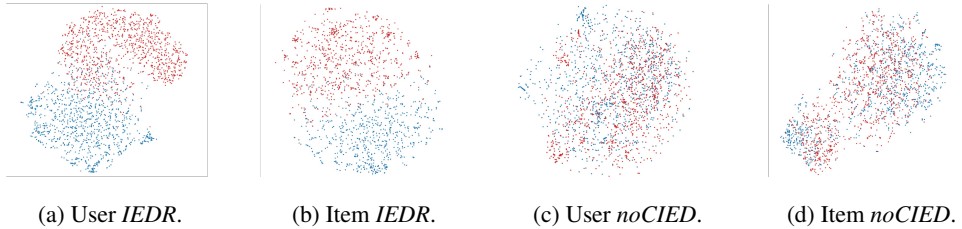

| (a) User *IEDR*. | (b) Item *IEDR*. | (c) User *noCIED*. | (d) Item *noCIED*. |

Figure 3: Visualization of the learned intrinsic and extrinsic representations with t-SNE for the Frappe dataset. The blue dots are the intrinsic representations, and the red dots are the extrinsic representations.

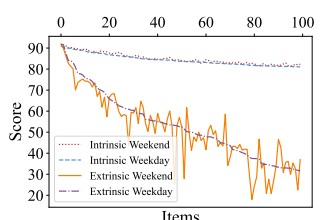

Figure 4: A user's top 100 intrinsic and extrinsic scores in different contexts (Weekend vs. Weekday).

Table 3: Items (in category) of the highest intrinsic and extrinsic scores for different users in Weekday.

| | User1 | | User2 | |
|---|---|---|---|---|
| Rank | Intrinsic | Extrinsic | Intrinsic | Extrinsic |
| 1 | Photography | Tool | Cards&Casino | Communication |
| 2 | Sports | Communication | Productivity | Tool |
| 3 | Health&Fitness | Media&Video | Cards&Casino | News&Magazines |
| 4 | Tools | Personalization | Sports Games | Tool |
| 5 | Health&Fitness | Communication | Brain&Puzzle | Communication |
| 6 | Personalization | Casual | Communication | News&Magazines |
| 7 | Personalization | Music&Audio | Tools | Personalization |
| 8 | Communication | News&Magazines | Sports | Media&Video |
| 9 | Personalization | Communication | Arcade&Action | Tool |
| 10 | Health&Fitness | Travel&Local | Tools | Communication |

top 100 items. The results are demonstrated in Figure 4. Note that the top 100 items for intrinsic and extrinsic factors are different. According to Figure 4, from weekday to weekend, the extrinsic scores vary a lot, while the intrinsic scores remain invariant. These observations demonstrate that, in different contexts, the user has different intrinsic factors, as well as consistent intrinsic factors.

Then, we illustrate how intrinsic and extrinsic factors may have different impacts on users' choices. Table 3 lists the categories of the items with the 10 highest intrinsic/extrinsic scores for two users, respectively. we observe that users have individual intrinsic interests that show their real hobbits in personal time, e.g., *User1* prefers sports and fitness apps while *User2* prefers gaming apps. On the other hand, extrinsic factors give a higher rank to the items based on the contexts (Workday), e.g., Tool (Google Search) and Communication (Gmail) rank highest in *User1*'s extrinsic scores.

**Remark.** In addition to the above experiments, we further evaluate our model by 1) running our model on other context/feature modeling methods (Appendix G.1), 2) comparing the proposed model with a naive baseline in learning intrinsic factor representations (Appendix G.3), and 3) studying how the hyperparameter settings influence the performance (Appendix G.6).

## 6 CONCLUSION

Capturing accurate intrinsic and extrinsic factors from contexts is an essential research topic in recommender systems. Focusing on the problem of existing studies that either neglect the factor learning, or learn the factors from only one specific context, we propose the intrinsic-extrinsic disentangled recommendation (IEDR) model. This generic model effectively learns intrinsic and extrinsic factors from various contexts for a more accurate recommendation. IEDR comprises a context-invariant contrastive learning component, and a mutual information minimization-based disentangling component to ensure the success of the factor learning. Extensive experiments prove our model's ability to learn intrinsic and extrinsic factors and leverage the learned factors for more accurate recommendation prediction. Following this work, we may discover other types of factors that can be considered besides the intrinsic and extrinsic ones, and learn more fine-grained intrinsic and extrinsic factors (e.g., multiple intrinsic factors).

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

# A PRELIMINARY

## A.1 STATISTICAL INTERACTION GRAPH NETWORK (SIGN)

The statistical interaction graph network (SIGN) (Su et al., 2021) explicitly models feature interactions through a graph neural network. IEDR, learns a user representation $\boldsymbol{u}$ (as well as a context representation, and an item representation) using the SIGN-based model $f_u(u)$, where $u = \{z_1^u, z_2^u, ..., z_p^u\}$, $f_u$ regards the user $u$ as a user graph $\mathcal{G}(\mathcal{V}, \mathcal{E})$. In this representtation, $\mathcal{V} = u$ is the node set that each feature $z_i^u$ is a node, and $\mathcal{E}$ is the edge set containing all the combinations of pairwise feature interactions, with each feature interaction $\langle z_i^u, z_j^u \rangle$ being an edge linking to corresponding nodes. Accordingly, the user representation learning becomes a graph learning problem.

In SIGN, first, each feature $z_i^u$ is mapped into a feature embedding $\boldsymbol{z}_i^u \in \mathbb{R}^d$ of $d$ dimensions as the node embedding. The embeddings are first randomly initialized and are updated through training. Then, SIGN learns the user graph using the function $f_u$:

$$f_u(\mathcal{G}) = \phi(\{\psi(\{e_{ij}h(\boldsymbol{z}_i^u, \boldsymbol{z}_j^u)\}_{j \in \mathcal{V}})\}_{i \in \mathcal{V}}), \tag{5}$$

where $\phi$ and $\psi$ are aggregation functions (e.g., element-wise mean), $h(\cdot) : \mathbb{R}^{2 \times d} \to \mathbb{R}^d$ is an MLP that models each feature interaction, $e_{ij} \in \{0, 1\}$ is the edge indicator (since we use all pairwise feature interactions, $e_{ij} = 1$ for all edges).

$f_u$ outputs the modeled user representation $\boldsymbol{u} \in \mathbb{R}^d$ of $d$ dimensions.

## A.2 VARIATIONAL CONTRASTIVE LOG-RATIO UPPER BOUND (VCLUB) OF MUTUAL INFORMATION

Suppose we aim to learn an intrinsic representation $\boldsymbol{o}_{in}^u$ and an extrinsic representation $\boldsymbol{o}_{ex}^u$, where their mutual information is minimized. In the vCLUB method (Cheng et al., 2020), a variational distribution $q_1^u(\boldsymbol{o}_{ex}^u | \boldsymbol{o}_{in}^u; \boldsymbol{\theta}_1^u)$ of parameter $\boldsymbol{\theta}_1^u$ (e.g., an MLP) is proposed to predict the extrinsic factor given the intrinsic factor. Then, the vCLUB-based mutual information upper bound can be derived as:

$$\mathcal{I}_{\text{vCLUB}}(\boldsymbol{o}_{in}^u; \boldsymbol{o}_{ex}^u) := \mathbb{E}_{p(\boldsymbol{o}_{in}^u, \boldsymbol{o}_{ex}^u)}[\log q_1^u(\boldsymbol{o}_{ex}^u | \boldsymbol{o}_{in}^u)] - \mathbb{E}_{p(\boldsymbol{o}_{in}^u)p(\boldsymbol{o}_{ex}^u)}[\log q_1^u(\boldsymbol{o}_{ex}^u | \boldsymbol{o}_{in}^u)], \tag{6}$$

where $p(\boldsymbol{o}_{in}^u, \boldsymbol{o}_{ex}^u)$ is the joint distribution and $p(\boldsymbol{o}_{in}^u)p(\boldsymbol{o}_{ex}^u)$ are is the marginal distribution.

vCLUB performs mutual information estimation and minimization in two steps iteratively. In the first step, to ensure Equation (6) holds as the upper bound, $\boldsymbol{\theta}_1^u$ is trained to make the log-likelihood function $\mathcal{L}_{appr}(u, c) := \frac{1}{N} \sum_{i=1}^N \log q_1^u((\boldsymbol{o}_{ex}^u)_i | (\boldsymbol{o}_{in}^u)_i)$ maximized (Theorem 3.2 of (Cheng et al., 2020)). In the second step, $\boldsymbol{\theta}_1^u$ is frozen, and other parameters (e.g., the parameters to generate $\boldsymbol{o}_{in}^u$ and $\boldsymbol{o}_{ex}^u$) are trained to minimize $\mathcal{I}_{\text{vCLUB}}(\boldsymbol{o}_{in}^u; \boldsymbol{o}_{ex}^u)$ so that the mutual information is minimized.

# B PROOF OF THEOREM 1

*Proof.* Since the mutual information is not explicitly intractable, we approximate the right side of Equation (4) with a lower bound (i.e., MINE (Belghazi et al., 2018)) and an upper bound (i.e., CLUB (Cheng et al., 2020)) of mutual information, respectively. More formally,

$$\mathcal{I}(\boldsymbol{o}_{in}^u, \boldsymbol{u}) \geq \mathcal{I}_{MINE}(\boldsymbol{o}_{in}^u, \boldsymbol{u}) := \mathbb{E}_{p(\boldsymbol{o}_{in}^u, \boldsymbol{u})}\left[\log p(\boldsymbol{o}_{in}^u, \boldsymbol{u})\right] - \log \mathbb{E}_{p(\boldsymbol{o}_{in}^u)p(\boldsymbol{u})}\left[p(\boldsymbol{o}_{in}^u, \boldsymbol{u})\right]; \tag{7}$$

$$\mathcal{I}(\boldsymbol{o}_{in}^u, \boldsymbol{c}) \leq \mathcal{I}_{CLUB}(\boldsymbol{o}_{in}^u, \boldsymbol{c}) := \mathbb{E}_{p(\boldsymbol{o}_{in}^u, \boldsymbol{c})}\left[\log p(\boldsymbol{o}_{in}^u | \boldsymbol{c})\right] - \mathbb{E}_{p(\boldsymbol{o}_{in}^u)p(\boldsymbol{c})}\left[\log p(\boldsymbol{o}_{in}^u | \boldsymbol{c})\right]. \tag{8}$$

With the approximated terms above, proving Equation. (4) turns to verify:

$$\arg\min \sum_{i=1}^N \mathcal{L}_{\text{CICL}}(u_i, c_i) = \arg\max \left(\mathcal{I}_{MINE}(\boldsymbol{o}_{in}^u, \boldsymbol{u}) - \mathcal{I}_{CLUB}(\boldsymbol{o}_{in}^u, \boldsymbol{c})\right). \tag{9}$$

By minimizing $\mathcal{L}_{\text{CICL}}$, we aim to make $(\boldsymbol{o}_{in}^u)_{ii}$ similar to $(\boldsymbol{o}_{in}^u)_{ij}$. This procedure can be interpreted in probability as: increasing the probability of $f_{ie}^u(\boldsymbol{u}_i, \boldsymbol{c}_j)$ to predict $(\boldsymbol{o}_{in}^u)_{ii}$. Therefore, maximizing the $\exp(\text{sim}((\boldsymbol{o}_{in}^u)_{ii}, (\boldsymbol{o}_{in}^u)_{ij})/\tau)$ in Equation (1) is equivalent to maximizing $p((\boldsymbol{o}_{in}^u)_{ii} | \boldsymbol{u}_i, \boldsymbol{c}_j)$

$(\exp(\cdot)$ is monotone increasing so that does not influence the conclusion). Similarly, minimizing $\exp(\mathrm{sim}((\boldsymbol{o}_{in}^u)_{ii}, (\boldsymbol{o}_{in}^u)_{\ell i})/\tau)$ is equivalent to minimizing $p((\boldsymbol{o}_{in}^u)_{ii}|\boldsymbol{u}_\ell, \boldsymbol{c}_i)$. Therefore, we have

$$-\sum_{i=1}^N \mathcal{L}_{\mathrm{CICL}}(u_i, c_i)$$

$$=\sum_{i=1}^N \log \frac{\exp(\mathrm{sim}((\boldsymbol{o}_{in}^u)_{ii}, (\boldsymbol{o}_{in}^u)_{ij})/\tau)}{\sum_{u_\ell \in \mathcal{U}} \exp(\mathrm{sim}((\boldsymbol{o}_{in}^u)_{ii}, (\boldsymbol{o}_{in}^u)_{\ell i})/\tau)}$$

$$=\sum_{i=1}^N \log[\exp(\mathrm{sim}((\boldsymbol{o}_{in}^u)_{ii}, (\boldsymbol{o}_{in}^u)_{ij})/\tau)] - \sum_{i=1}^N \log[\sum_{u_\ell \in \mathcal{U}} \exp(\mathrm{sim}((\boldsymbol{o}_{in}^u)_{ii}, (\boldsymbol{o}_{in}^u)_{\ell i})/\tau)]$$

$$=\sum_{i=1}^N \log[p((\boldsymbol{o}_{in}^u)_{ii}|\boldsymbol{u}_i, \boldsymbol{c}_j)] - \sum_{i=1}^N \log[\sum_{u_\ell \in \mathcal{U}} p((\boldsymbol{o}_{in}^u)_{ii}|\boldsymbol{u}_\ell, \boldsymbol{c}_i)].$$

Equation (1) only samples one context $c_j$ for each data point. However, during the training, all contexts in $\mathcal{C}$ are expected to be sampled. If we count all contexts, we have

$$\sum_{i=1}^N \log[p((\boldsymbol{o}_{in}^u)_{ii}|\boldsymbol{u}_i, \boldsymbol{c}_j)] - \sum_{i=1}^N \log[\sum_{u_\ell \in \mathcal{U}} p((\boldsymbol{o}_{in}^u)_{ii}|\boldsymbol{u}_\ell, \boldsymbol{c}_i)]$$

$$=\sum_{i=1}^N \sum_{c_j \in \mathcal{C}} \log[p((\boldsymbol{o}_{in}^u)_{ii}|\boldsymbol{u}_i, \boldsymbol{c}_j)] - \sum_{i=1}^N \log[\sum_{u_\ell \in \mathcal{U}} p((\boldsymbol{o}_{in}^u)_{ii}|\boldsymbol{u}_\ell, \boldsymbol{c}_i)] \qquad (10)$$

$$=\mathbb{E}_{p(\boldsymbol{o}_{in}^u, \boldsymbol{u})p(\boldsymbol{c})}[\log p(\boldsymbol{o}_{in}^u|\boldsymbol{u}, \boldsymbol{c}) - \mathbb{E}_{p(\boldsymbol{o}_{in}^u, \boldsymbol{c})} \log \mathbb{E}_{p(\boldsymbol{u})}[p(\boldsymbol{o}_{in}^u|\boldsymbol{u}, \boldsymbol{c})]$$

Equation (10) is the probability form of the objective function of the context-invariant counteractive learning component (Equation (1)). Equation (10) maximizes the likelihood $p(\boldsymbol{o}_{in}^u|\boldsymbol{u}, \boldsymbol{c})$ given the joint distribution of users and intrinsic factors, with the marginal distribution of contexts. Meanwhile, it minimizes the likelihood $p(\boldsymbol{o}_{in}^u|\boldsymbol{u}, \boldsymbol{c})$ given the joint distribution of contexts and intrinsic factors, with the marginal distribution of user.[2]

From Equation (10), we further have:

$$\mathbb{E}_{p(\boldsymbol{o}_{in}^u, \boldsymbol{u})p(\boldsymbol{c})}[\log p(\boldsymbol{o}_{in}^u|\boldsymbol{u}, \boldsymbol{c})] - \mathbb{E}_{p(\boldsymbol{o}_{in}^u, \boldsymbol{c})} \log \mathbb{E}_{p(\boldsymbol{u})}[p(\boldsymbol{o}_{in}^u|\boldsymbol{u}, \boldsymbol{c})]$$

$$\overset{(a)}{=} \mathbb{E}_{p(\boldsymbol{o}_{in}^u, \boldsymbol{u})p(\boldsymbol{c})}[\log p(\boldsymbol{o}_{in}^u|\boldsymbol{u}, \boldsymbol{c})] - \mathbb{E}_{p(\boldsymbol{o}_{in}^u, \boldsymbol{c})p(\boldsymbol{u})}[\log p(\boldsymbol{o}_{in}^u|\boldsymbol{u}, \boldsymbol{c})]$$

$$= \mathbb{E}_{p(\boldsymbol{o}_{in}^u, \boldsymbol{u})p(\boldsymbol{c})}[\log p(\boldsymbol{o}_{in}^u|\boldsymbol{u}, \boldsymbol{c})] - \mathbb{E}_{p(\boldsymbol{o}_{in}^u, \boldsymbol{c})p(\boldsymbol{u})}[\log p(\boldsymbol{o}_{in}^u|\boldsymbol{u}, \boldsymbol{c})] + \left(\mathbb{E}_{p(\boldsymbol{u})}[\log p(\boldsymbol{u})] - \mathbb{E}_{p(\boldsymbol{u})}[\log p(\boldsymbol{u})]\right)$$

$$= \mathbb{E}_{p(\boldsymbol{o}_{in}^u, \boldsymbol{u})p(\boldsymbol{c})}[\log p(\boldsymbol{o}_{in}^u|\boldsymbol{u}, \boldsymbol{c})p(\boldsymbol{u})] - \mathbb{E}_{p(\boldsymbol{o}_{in}^u, \boldsymbol{c})p(\boldsymbol{u})}[\log p(\boldsymbol{o}_{in}^u|\boldsymbol{u}, \boldsymbol{c})] - \mathbb{E}_{p(\boldsymbol{u})}[\log p(\boldsymbol{u})]$$

$$= \mathbb{E}_{p(\boldsymbol{o}_{in}^u, \boldsymbol{u})p(\boldsymbol{c})}[\log p(\boldsymbol{o}_{in}^u, \boldsymbol{u}|\boldsymbol{c})] - \mathbb{E}_{p(\boldsymbol{o}_{in}^u, \boldsymbol{c})p(\boldsymbol{u})}[\log p(\boldsymbol{o}_{in}^u|\boldsymbol{u}, \boldsymbol{c})] - \mathbb{E}_{p(\boldsymbol{u})}[\log p(\boldsymbol{u})]$$

$$= \mathbb{E}_{p(\boldsymbol{o}_{in}^u, \boldsymbol{u})p(\boldsymbol{c})}[\log p(\boldsymbol{o}_{in}^u, \boldsymbol{u}|\boldsymbol{c})] - \mathbb{E}_{p(\boldsymbol{o}_{in}^u, \boldsymbol{c})p(\boldsymbol{u})}[\log p(\boldsymbol{o}_{in}^u|\boldsymbol{u}, \boldsymbol{c})] - \mathbb{E}_{p(\boldsymbol{u})}[\log p(\boldsymbol{u})]$$

$$\quad + \left(\mathbb{E}_{p(\boldsymbol{o}_{in}^u)p(\boldsymbol{u})p(\boldsymbol{c})}[\log p(\boldsymbol{o}_{in}^u|\boldsymbol{u}, \boldsymbol{c})] - \mathbb{E}_{p(\boldsymbol{o}_{in}^u)p(\boldsymbol{u})p(\boldsymbol{c})}[\log p(\boldsymbol{o}_{in}^u|\boldsymbol{u}, \boldsymbol{c})]\right)$$

$$= \mathbb{E}_{p(\boldsymbol{o}_{in}^u, \boldsymbol{u})p(\boldsymbol{c})}[\log p(\boldsymbol{o}_{in}^u, \boldsymbol{u}|\boldsymbol{c})] - \mathbb{E}_{p(\boldsymbol{o}_{in}^u)p(\boldsymbol{u})p(\boldsymbol{c})}[\log p(\boldsymbol{o}_{in}^u|\boldsymbol{u}, \boldsymbol{c})] - \mathbb{E}_{p(\boldsymbol{u})}[\log p(\boldsymbol{u})]$$

$$\quad - \mathbb{E}_{p(\boldsymbol{o}_{in}^u, \boldsymbol{c})p(\boldsymbol{u})}[\log p(\boldsymbol{o}_{in}^u|\boldsymbol{u}, \boldsymbol{c})] + \mathbb{E}_{p(\boldsymbol{o}_{in}^u)p(\boldsymbol{u})p(\boldsymbol{c})}[\log p(\boldsymbol{o}_{in}^u|\boldsymbol{u}, \boldsymbol{c})]$$

$$= \mathbb{E}_{p(\boldsymbol{o}_{in}^u, \boldsymbol{u})p(\boldsymbol{c})}[\log p(\boldsymbol{o}_{in}^u, \boldsymbol{u}|\boldsymbol{c})] - \mathbb{E}_{p(\boldsymbol{o}_{in}^u)p(\boldsymbol{u})p(\boldsymbol{c})}[\log p(\boldsymbol{o}_{in}^u, \boldsymbol{u}|\boldsymbol{c})]$$

$$\quad - \left(\mathbb{E}_{p(\boldsymbol{o}_{in}^u, \boldsymbol{c})p(\boldsymbol{u})}[\log p(\boldsymbol{o}_{in}^u|\boldsymbol{u}, \boldsymbol{c})] - \mathbb{E}_{p(\boldsymbol{o}_{in}^u)p(\boldsymbol{u})p(\boldsymbol{c})}[\log p(\boldsymbol{o}_{in}^u|\boldsymbol{u}, \boldsymbol{c})]\right)$$

$$= \mathbb{E}_{p(\boldsymbol{c})} \left(\mathbb{E}_{p(\boldsymbol{o}_{in}^u, \boldsymbol{u})}[\log p(\boldsymbol{o}_{in}^u, \boldsymbol{u}|\boldsymbol{c})] - \mathbb{E}_{p(\boldsymbol{o}_{in}^u)p(\boldsymbol{u})}[\log p(\boldsymbol{o}_{in}^u, \boldsymbol{u}|\boldsymbol{c})]\right)$$

$$\quad - \mathbb{E}_{p(\boldsymbol{u})} \left(\mathbb{E}_{p(\boldsymbol{o}_{in}^u, \boldsymbol{c})}[\log p(\boldsymbol{o}_{in}^u|\boldsymbol{u}, \boldsymbol{c})] - \mathbb{E}_{p(\boldsymbol{o}_{in}^u)p(\boldsymbol{c})}[\log p(\boldsymbol{o}_{in}^u|\boldsymbol{u}, \boldsymbol{c})]\right).$$

$$(11)$$

---

[2]Note that only if $f_{ie}^u(\boldsymbol{u}, \boldsymbol{c})$ is a many-to-one (or one-to-one) mapping then Equation (10) and Equation (1) will be equivalent. Otherwise, given a sample pair $(\boldsymbol{u}, \boldsymbol{c})$, $f_{ie}^u(\boldsymbol{u}, \boldsymbol{c})$ may have different $\boldsymbol{o}_{in}^u$ outputs (i.e., one-to-many). In this situation, the first term of Equation (10) cannot guarantee that the same user with different context will have the same intrinsic factor (i.e., they may have various intrinsic factor representations while still meet the objective of the first term of Equation (10)). We use an MLP as $f_{ie}^u(\boldsymbol{u}, \boldsymbol{c})$, which is a many-to-one mapping function. Therefore, we can ensure the equivalence between Equation (10) and Equation (1).

(a): In the second term, pushing the log inside the expectation dose not change the minimizer.

Comparing Equation (7) and the first term of Equation (11), they both act like classifiers whose objectives maximize the expected log-ratio of the joint distribution over the product of marginal distributions (Hjelm et al., 2019). Therefore, maximizing this term in Equation (11) will have the same effect to maximizing Equation (7). We can interpret the first term of Equation (11) as maximizing the mutual information between users and the corresponding intrinsic factor, conditioned on a given context. Similarly, maximizing the negative of the second term of Equation (11) will have the same effect of minimizing Equation (8), which can be interpreted as minimizing the mutual information between contexts and the corresponding intrinsic factors, conditioned on a given users.

Therefore, we can conclude that:

$$\arg\min \sum_{(u_i, v_i, c_i) \in \mathcal{D}} \mathcal{L}_{\text{CICL}}(u_i, c_i) = \arg\max \mathcal{I}_{MINE}(\boldsymbol{o}_{in}^u, \boldsymbol{u}) - \mathcal{I}_{CLUB}(\boldsymbol{o}_{in}^u, \boldsymbol{c}).$$

$\square$

## C PREVENTING THE TRIVIAL SOLUTION OF CIED

The two components in the CIED module, the contrastive learning component and the disentangling component, jointly ensure the success of the intrinsic and extrinsic factor representation learning. However, CIED may fall into a trivial solution: $f_{ie}^u(\boldsymbol{u}, \boldsymbol{c})$ maps $u$ to $\boldsymbol{o}_{in}^u$ without considering $c$, and maps $c$ to $\boldsymbol{o}_{ex}^u$ without considering $u$. Although this trivial solution minimizes $\mathcal{L}_{\text{CICL}}(u, c)$ and $\mathcal{L}_{Dis}(u, c)$, $\boldsymbol{o}_{in}^u$ (resp. $\boldsymbol{o}_{ex}^u$) is not the intrinsic (resp. extrinsic) factor, but just a mapping of the user information (resp. context information). We prove that this trivial solution can be avoided by setting $f_{ie}^u(\boldsymbol{u}, \boldsymbol{c})$ as a *non-linear* function, leading $\boldsymbol{u}$ and $\boldsymbol{c}$ statistically interacted.

**Statistical Interaction** We start with introducing the statistical interaction (or non-additive interaction), which ensures a joint influence of several variables on an output variable is not additive (Tsang et al., 2018). Based on Sorokina et al. (2008), $F(\boldsymbol{X})$ shows statistical interaction between variables $x_i$ and $x_j$ if $\forall f_{\backslash i}, f_{\backslash j}, F(\boldsymbol{X})$ **cannot** be expressed as:

$$F(\boldsymbol{X}) \neq f_{\backslash i}(x_1, \ldots, x_{i-1}, x_{i+1}, \ldots, x_n) + f_{\backslash j}(x_1, \ldots, x_{j-1}, x_{j+1}, \ldots, x_n). \qquad (12)$$

More generally, if using $\boldsymbol{v}_i \in \mathbb{R}^d$ to describe the $i$-th variable with a $d$-dimension vector (Rendle, 2010; Su et al., 2021), e.g., variable embedding, each variable can be described in a vector form $\boldsymbol{u}_i = x_i \boldsymbol{v}_i$. Then, we define the pairwise statistical interaction in vector form by changing the Equation (12) into:

$$F(\boldsymbol{X}) \neq f_{\backslash i}(\boldsymbol{u}_1, \ldots, \boldsymbol{u}_{i-1}, \boldsymbol{u}_{i+1}, \ldots, \boldsymbol{u}_n) + f_{\backslash j}(\boldsymbol{u}_1, \ldots, \boldsymbol{u}_{j-1}, \boldsymbol{u}_{j+1}, \ldots, \boldsymbol{u}_n). \qquad (13)$$

**Preventing the Trivial Solution** Based on the definition of statistical interaction, we can express the trivial solution as that $f_{ie}^u(\boldsymbol{u}, \boldsymbol{c})$ learns no statistical interaction between $\boldsymbol{u}$ and $\boldsymbol{c}$:

$$f_{ie}^u(\boldsymbol{u}, \boldsymbol{c}) = \lambda_1 f_1(\boldsymbol{u}) + \lambda_2 f_2(\boldsymbol{c}), \qquad (14)$$

where $f_1$ outputs $\boldsymbol{o}_{in}^u$, $f_2$ outputs $\boldsymbol{o}_{ex}^u$, and $\lambda$ are weight scalars.

To prevent the trivial solution, we need to ensure that function $f_{ie}^u(\boldsymbol{u}, \boldsymbol{c})$ cannot be modeled in the form of Equation (14). Therefore, if $\boldsymbol{u}$ and $\boldsymbol{c}$ are modeled as a statistical interaction in $f_{ie}^u(\boldsymbol{u}, \boldsymbol{c})$, the trivial solution can be prevented. Since $f_{ie}^u(\boldsymbol{u}, \boldsymbol{c})$ only takes $\boldsymbol{u}$ and $\boldsymbol{c}$ as inputs, we just need $f_{ie}^u$ to be a non-additive model. That is, $f_{ie}^u(\boldsymbol{u}, \boldsymbol{c})$ should contain a third term $f_3(\boldsymbol{u}, \boldsymbol{c})$:

$$f_{ie}^u(\boldsymbol{u}, \boldsymbol{c}) = \lambda_1 f_1(\boldsymbol{u}) + \lambda_2 f_2(\boldsymbol{c}) + \lambda_3 f_3(\boldsymbol{u}, \boldsymbol{c}), \qquad (15)$$

where $f_3$ is a non-additive model and $\lambda_3 \neq 0$.

Therefore, in the optimized situation, $\boldsymbol{o}_{in}^u = \lambda_1 f_1(\boldsymbol{u})$ learns part of the information from users that do not interact with context information. $\boldsymbol{o}_{ex}^u = \lambda_2 f_2(\boldsymbol{c}) + \lambda_3 f_3(\boldsymbol{u}, \boldsymbol{c})$ learns the context information ($f_2(\boldsymbol{c})$) and the information that changes given different contexts ($f_3(\boldsymbol{u}, \boldsymbol{c})$).

In Appendix G.2, we empirically analyze how the trivial solution will influence the prediction performance.

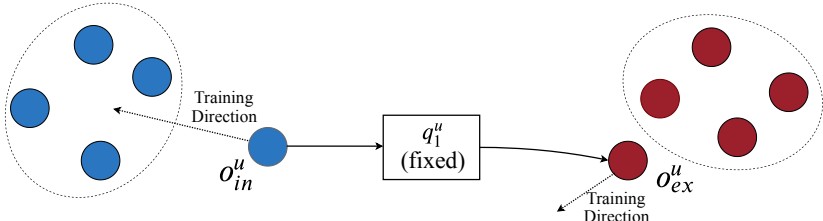

Figure 5: An illustrative example demonstrating the potential problem of asymmetric learning in vCLUB. The blue circles are intrinsic representations, and the red circles are extrinsic representations. The dotted arrows are the directions that *vCLUB* will push $\boldsymbol{o}_{in}^u$ and $\boldsymbol{o}_{ex}^u$ to move toward in their space.

## D   POTENTIAL PROBLEMS OF THE ASYMMETRIC VCLUB METHOD

The vCLUB-based mutual information minimization method proposed in (Cheng et al., 2020) is an asymmetric method. Appendix A.2 gives an introduction about how vCLUB method performs mutual information minimization. In this section we explain the possible reason that *vCLUB* is less robust and perform worse than our proposed bidirectional vCLUB method (*BiDis*).

If we directly apply vCLUB to our disentangling component, the parameter $\boldsymbol{\theta}_1^u$ of a variational distribution $q_1^u(\boldsymbol{o}_{ex}^u | \boldsymbol{o}_{in}^u; \boldsymbol{\theta}_1^u)$ will be trained to approach the vCLUB-based upper bound in Equation (6) (Step 1). Then, $\boldsymbol{\theta}_1^u$ is frozen, and $\boldsymbol{o}_{ex}^u, \boldsymbol{o}_{in}^u$ are trained to minimize $\mathcal{I}(\boldsymbol{o}_{in}^u; \boldsymbol{o}_{ex}^u)$ via minimizing the upper bound $\mathcal{I}_{\text{vCLUB}}(\boldsymbol{o}_{in}^u; \boldsymbol{o}_{ex}^u)$ (Step 2). However, this way of minimizing mutual information may result in an unexpected outcome: the mutual information may be minimized via making $\boldsymbol{o}_{in}^u$ contain as less information as possible. To better illustrate the possible outcome, we design $q_1^u$ as a linear function and is well trained in Step 1 to ensure Equation (6) is an upper bound of $\mathcal{I}(\boldsymbol{o}_{in}^u; \boldsymbol{o}_{ex}^u)$. Figure 5 shows how the unexpected result may occur. In Step 2, $\boldsymbol{o}_{ex}^u, \boldsymbol{o}_{in}^u$ will be trained to minimize Equation (6). To achieve this goal, it ensures $q_1^u$ cannot predict $\boldsymbol{o}_{ex}^u$ given the corresponding $\boldsymbol{o}_{in}^u$ from the joint distribution (the first term of Equation (6)), and at the same time ensures the output of $q_1^u$ is similar to the other $\boldsymbol{o}_{ex}^u$'s from the marginal distribution (the second term of Equation (6)).

From $\boldsymbol{o}_{in}^u$ perspective (blue circles), the goal can be achieved by pushing the $\boldsymbol{o}_{in}^u$ to move from its original position (optimizing the first term of Equation (6)), and move towards the mean of the other $\boldsymbol{o}_{in}^u$'s (optimizing the second term of Equation (6)). From $\boldsymbol{o}_{ex}^u$ perspective (red circles), the goal can be achieved by pushing the $\boldsymbol{o}_{ex}^u$ away from its original position (optimizing the first term of Equation (6)) and the the mean of the other $\boldsymbol{o}_{ex}^u$'s (optimizing the second term of Equation (6)).

This clusters all the $\boldsymbol{o}_{in}^u$'s together, making $\boldsymbol{o}_{in}^u$'s contain less information, while all the $\boldsymbol{o}_{ex}^u$'s try to split away from each other, making $\boldsymbol{o}_{ex}^u$'s contain more information. The mutual information minimization procedure is like "transfering" the information from $\boldsymbol{o}_{in}^u$'s to $\boldsymbol{o}_{ex}^u$'s, which is not what we expect. *BiDis*, however, is a symmetric disentangling method on $\boldsymbol{o}_{in}^u$'s and $\boldsymbol{o}_{ex}^u$'s so that will not result in this issue. This may be the reason that *vCLUB* performs worse and less robust than our proposed symmetrical disentangling component.

## E   TIME COMPLEXITY ANALYSIS

Briefly speaking, the time complexity of the whole model is comparable to feature interaction-based recommender systems (e.g., AutoInt, SIGN). The overhead of the alternative optimizing procedure for the disentanglement component is marginal in the whole optimizing procedure.

The most time-consuming computations are the feature interaction learning to get user, item, and context representations, which need to conduct interaction modeling on every pair of feature interactions. This procedure has also been done on other feature interaction-based models, therefore, the time complexity of the proposed module is comparable with those methods.

Our model takes additional computations on the context-invariant contrastive learning (CICL) component and the disentangling component:

For the CICL component, we do not need to perform the feature interaction modeling again, but reuse the generated user/item/context representations, which saves the majority of the overhead. We only need to perform $f_{ie}$ (L+1) times, where L is the number of negative samples. Since $f_{ie}$ is a one-hidden layer (with 128 hidden units) MLP, the overhead is marginal.

For the disentangling component, we also reuse the generated user/item/context representations. As we discussed in the paper, we use a two-step learning policy to train our model. Regarding to the reviewer's main concern, the first step in the two-step learning actually takes very little overhead. This is because this step only tries to optimize the parameters of the functions $q_1$ and $q_2$ (Eq.2), which are two MLPs with one hidden layer. For each data sample, we only run $q_1$ and $q_2$ one time using $O_{in}$ and $O_{ex}$.

In summary, since all of the computations above do not need to perform feature interaction modeling (the most time-consuming procedure in all feature interaction-based models), the small imposed overhead is acceptable considering the effectiveness of our model in capturing accurate intrinsic and extrinsic factors.

## F  EXPERIMENTAL SETTING

**Datasets**  We evaluate our models in two scenarios with various contexts: a mobile app recommendation and a restaurant recommendation. In the mobile app recommendation, we use the Frappe (Baltrunas et al., 2015) dataset that records mobile app usage logs. Each data sample logs users' app usage in a certain context (e.g., weather, time, location). In addition, some relevant properties of the apps are also captured (e.g., category, developer). In the restaurant recommendation, we use the Yelp dataset (Wu et al., 2022). It records users' reviews on local restaurants. Due to the fact that a user usually goes to restaurants in the same city, geographic isolation appears in the dataset. Therefore, we select the records in New York City. We regard each record as a data sample that the user has been to the restaurant. We leverage the user/item features and context features (e.g., day of the week) to predict whether a user will go to a given restaurant in a specific context. We also evaluate our model in two Amazon datasets (Movies and CDs) (McAuley et al., 2015) that have been used in sequential recommendation tasks (Yu et al., 2019b). The datasets contain user-item interactions with timestamps. For the sequential recommendation, we use the same IEDR model structure as that for the Frappe and Yelp datasets, but modify the data input to fit our model. More specifically, we do not directly learn behavior sequences, but consider each behavior as a data sample with time context information. That is, we consider the bucketed timestamp of each user behavior as a time context (we consider one month as a categorized time context). Therefore, behaviors in the same time interval have the same time context, indicating that these behaviors share some similar short-term (extrinsic) interests (e.g., item popularity).

For each dataset, only users that have more than 5 records (Frappe and Yelp) and more than 20 records (Movies and CDs) are chosen. We choose the last record of each user for testing, and the second last record of each user for validation. The rest of the records are used for training. Each of these data sample is considered as a positive sample ($y = 1$). In addition, for each positive data sample in the training set, we randomly choose 2 items (but keep the user and contexts) as negative samples ($y = 0$), meaning the user did not select the 2 items in that context. For each test/validation data sample, we randomly choose 99 items as negative samples to ensure a more robust evaluation. The statistics of the datasets are shown in Table 4.

Table 4: Dataset statistics. "Count" refers to the number of users/items, "Features" represent the number of different features (for User and Item, the number of features excludes the user/item ids).

| Datasets | Data Samples | | | User | | Item | | Context |
| --- | --- | --- | --- | --- | --- | --- | --- | --- |
| | Train | Valid | Test | Count | Features | Count | Features | Features |
| Frappe | 282,426 | 69,500 | 69,500 | 695 | 0 | 4,082 | 2,892 | 318 |
| Yelp | 518,208 | 633,600 | 633,600 | 6,336 | 24 | 12,902 | 66 | 13,034 |
| Movies | 2,305,362 | 39,663 | 1,322,100 | 13,221 | 0 | 49,189 | 161 | 193 |
| CDs | 879,030 | 16,392 | 546,400 | 5,464 | 0 | 16,184 | 209 | 195 |

**Baseline methods**  IEDR models the feature interactions of users, items, and contexts. Therefore, we compare our model with competitive feature interaction-based recommendation methods. The methods include attentional factorization machine (AFM) (Xiao et al., 2017), neural factorization machine (NFM) (He & Chua, 2017), self-attention-based feature interaction model (AutoInt) (Song et al., 2019), deep factorization machine (DeepFM) (Guo et al., 2017), wide & deep model (WDL) (Cheng et al., 2016), improved deep & cross network (DCNv2) (Wang et al., 2021), and input-aware factorization machine (IFM) (Yu et al., 2019a). We implement these methods using the DeepCTR package. The statistical interaction graph neural network (SIGN) (Su et al., 2021) is applied based on the released code. The above methods model all the factors in a unified representation without considering the factors that affect users' decisions. Meanwhile, we compare IEDR with the methods that learn implicit factors. They are disentangled variational auto-encoder for recommendation (DisRec) (Ma et al., 2019) and disentangled graph collaborative filtering (DGCF) (Wang et al., 2020). We implement these methods based on their released code. Note that since DisRec and DGCF models do not consider any feature, their task is to simply predict whether a user will select an item. IERD and other baseline models, however, consider the user-item interactions in specific contexts (a user's decision to select an item may be different in different contexts). For DisRec and DGCF, to prevent the test data samples from appearing in the training set, we remove the data samples from the training set that appear in the test set (with different contexts in other models). For a fair comparison, we set the factor number to 4 for DisRec and DGCF. For sequential recommendation baselines, we compare our model with the models that consider LS-term interests. They are session-based recommender system with recurrent neural networks (GRU4Rec) (Hidasi et al., 2016), Short-term and Long-term preference Integrated Recommender system (SLI-Rec) (Yu et al., 2019b), and Contrastive learning framework of Long and Short-term interests for Recommendation (CLSR) (Zheng et al., 2022). We use the same MLP structure for feature interaction modeling and the same embedding size for features as our IEDR model.

**Implementation details**  In IEDR, all the MLPs have the same hidden structure: one hidden layer of 128 dimensions and a ReLU activation after that. The input and output sizes of MLPs varies based on their needs. We set the embedding dimension to 32 for all the features. $f_{ie}$ is an MLP that outputs a 64-dimension vector, with the first 32 dimensions are the intrinsic factor representation and the last 32 dimensions are the extrinsic factor representation. For the second (dropout-based) negative context generating method in the context-invariant contrastive learning component, the dropout rate is set to 0.5. The number of negative pairs for contrastive learning is 40 for each data sample (note that the actual negative pairs will be doubled since both $(o_{in}^u)_{ii}$ and $(o_{in}^u)_{ij}$ will generate 40 negative pairs). The temperature $\tau$ is set to 0.5. In the disentangling component, $q_1$ and $q_2$ are MLPs that output vectors that have the same dimension of intrinsic/extrinsic factor representations. The number of negative samples of the bidirectional vCLUB-based method is 5 for each direction. We set $\lambda_1$ to 0.1 for the Frappe dataset and 0.01 for the Yelp dataset, and set $\lambda_2$ to 0.1 for both datasets. The $\lambda_1$ and $\lambda_2$ are both 0.01 for the Movies and the CDs datasets. We run all the experiments on a machine equipped with a CPU: Intel(R) Xeon(R) Platinum 8163 CPU @ 2.50GHz, and a GPU: Nvidia Tesla v100 GPU.

The model structure of IEDR and its variations used in the experiments are detailed in Table 5 and Table 6. Note that the component structures of variations are the same as the IEDR if not specified.

# G  ADDITIONAL EXPERIMENTAL RESULTS

In this section, we provide further experimental results that are not included in the main paper.

## G.1  OTHER FEATURE MODELING METHODS

In the RP module, although we use a SIGN-based method (Su et al., 2021) to learn user, item, and context features, the module can use any feature modeling method. Here, we use other methods to evaluate whether our model still performs well. Specifically, we run our model with the other three variations using different feature modeling methods: 1) averaging feature embeddings (*MEAN*); 2) adding an MLP on top of the averaged feature embedding (*MLP*); and 3) modeling and aggregating feature interactions through a Bi-interaction layer proposed in (He & Chua, 2017) (*BI*). The results are shown in Figure 6. We report the results of each variation with and without the CIED module.

Table 5: Implementation details of different variations on the recommendation prediction module. "-" represent the operation is the same as our original IEDR setting.

| Variation | Recommendation Prediction Module | |
| --- | --- | --- |
| | feature model[†] | $f_{ie}$ [‡] |
| IEDR | $\phi(\psi(\{MLP(\boldsymbol{z}_i^u \odot \boldsymbol{z}_j^u)\}_{j \in u}))_{i \in u} \to \boldsymbol{u}$ | $MLP(\boldsymbol{u} \circ \boldsymbol{c}) \to [\boldsymbol{o}_{in}^u, \boldsymbol{o}_{ex}^u]$ |
| AVG | $\psi(\boldsymbol{z}_i^u)_{i \in u} \to \boldsymbol{u}$ | - |
| MLP | $MLP(\psi(\boldsymbol{z}_i^u)_{i \in u}) \to \boldsymbol{u}$ | - |
| BI | $\psi(\boldsymbol{z}_i^u \odot \boldsymbol{z}_j^u)_{i,j \in u} \to \boldsymbol{u}$ | - |
| Linear | - | $\boldsymbol{W}[\boldsymbol{u}, \boldsymbol{c}] \to [\boldsymbol{o}_{in}^u, \boldsymbol{o}_{ex}^u]$ |
| Nonlinear | - | $\sigma(\boldsymbol{W}[\boldsymbol{u}, \boldsymbol{c}]) \to [\boldsymbol{o}_{in}^u, \boldsymbol{o}_{ex}^u]$ |
| $IEDR_{sp}$ | - | $MLP_1(\boldsymbol{u}) \to \boldsymbol{o}_{in}^u, MLP_2(\boldsymbol{u} \circ \boldsymbol{c}) \to \boldsymbol{o}_{ex}^u$ |

[†]Here we use user representation learning as an example. The item and context learning have the same structure. $\phi, \psi$ are both element-wise averaging functions and $\odot$ is the element-wise product.

[‡]Here we use user factor learning as an example. $\circ$ is a flexible operation to combine two vector, i.e., $\circ$ is element-wise product for the Frappe dataset, and element-wise summation for the Yelp dataset. $[\cdot, \cdot]$ is the concatenation operation. $\boldsymbol{W}$ is a linear transformation matrix, $\sigma$ is a ReLU activation.

Table 6: Implementation details of different variations of the contrastive intrinsic-extrinsic disentanglement module. "-" represents the operation is the same as our original IEDR setting. $\times$ represents the variation does not contain the component.

| Variation | Contrastive Intrinsic-Extrinsic Disentanglement Module | |
| --- | --- | --- |
| | Contrastive Learning Component[*] | Disentangling Component |
| IEDR | $positive\ sample$: $f_{ie}^u(\boldsymbol{u}_i, \boldsymbol{c}_j) \to (\boldsymbol{o}_{in}^u)_{ij}$, $negative\ sample$: $f_{ie}^u(\boldsymbol{u}_\ell, \boldsymbol{c}_i) \to (\boldsymbol{o}_{in}^u)_{\ell i}$, $f_{ie}^u(\boldsymbol{u}_\ell, \boldsymbol{c}_j) \to (\boldsymbol{o}_{in}^u)_{\ell j}$. $c_j = randChoice(NegGen1, NegGen2)$ | $MLP_{\theta_1}(\boldsymbol{o}_{in}^u) \to (\boldsymbol{o}_{ex}^u)'\ (q_1^u)$ $MLP_{\theta_2}(\boldsymbol{o}_{ex}^u) \to (\boldsymbol{o}_{in}^u)'\ (q_2^u)$ |
| noDis | - | $\times$ |
| noCL | $\times$ | - |
| noCIED | $\times$ | $\times$ |
| NegGen1 | $c_j$ is generated from $NegGen1$ | - |
| NegGen2 | $c_j$ is generated from $NegGen2$ | - |
| NegGen1&2 | - | - |
| vCLUB | - | $MLP_{\theta_1}(\boldsymbol{o}_{in}^u) \to (\boldsymbol{o}_{ex}^u)'\ (q_1^u)$ |
| BiDis | - | - |
| $IEDR_{sp}$ | $positive\ sample$: $dropout((\boldsymbol{o}_{in}^u)_i) \to (\boldsymbol{o}_{in}^u)^p$, $negative\ sample$: $dropout((\boldsymbol{o}_{in}^u)_\ell) \to (\boldsymbol{o}_{in}^u)^n$. | - |

[*]For $IEDR_{sp}$, the positive samples $(\boldsymbol{o}_{in}^u)^p$ are generated through a dropout of the intrinsic representation of the user, and the negative samples $(\boldsymbol{o}_{in}^u)^p$ are generated through a dropout of intrinsic representations of random users.

From this figure, we can see that when equipped with the CIED module, all feature modeling methods perform better than those without the module. It shows that our proposed CIED module can learn intrinsic and extrinsic factors for more accurate recommendation when different feature modeling methods are applied. Meanwhile, we can see that the feature modeling methods can impact the performance. *MEAN* is just a linear aggregation of features, resulting in the worst performance. Both *MLP* and *BI* have better feature modeling ability and hence have better performance than *MEAN*. The SIGN-based feature modeling (*SIGN*) is the state-of-the-art feature interaction modeling method and performs the best.

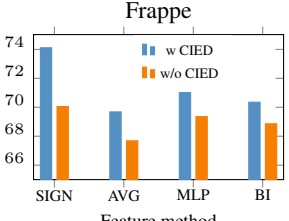 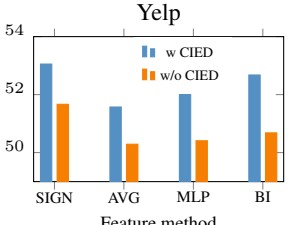

Figure 6: Model performance when equipped with different feature modeling methods.

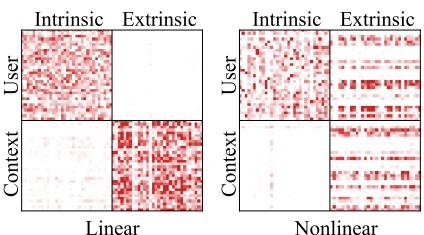 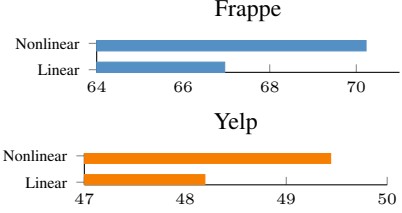

Figure 7: Visualization of $f_{ie}$ weights for the *Linear* and *Nonlinear* models.

Figure 8: Comparing the performance of the *Linear* and *Nonlinear* models on different datasets.

### G.2 FALLING INTO TRIVIAL SOLUTIONS

As discussed in Section C, our model may fall into a trivial solution if $f_{ie}^u(\boldsymbol{u}, \boldsymbol{c})$ is a linear mapping method. To evaluate how the trivial solution influences our model in learning the factors, we run our model with $f_{ie}$ being linear. Specifically, we concatenate $\boldsymbol{u}$ and $\boldsymbol{c}$ and feed them into an MLP without a hidden layer or activation (a linear mapping), making it easy to fall into the trivial solution. We call this variation *Linear*. Then, we avoid this by simply adding a nonlinear activation function (ReLU) activation after the linear mapping. We call this variation *Nonlinear*. Figure 7 shows the weight values of $f_{ie}$ of the two variations. The color shows the weights mapping from user/context representations to intrinsic/extrinsic representations. The darker the color, the larger the weight (more information of user/context is mapped into intrinsic/extrinsic representations). The figure shows that in the *Linear* variation, user information is largely mapped into intrinsic representation (user-intrinsic block) but not extrinsic representation (user-extrinsic block). Context information is largely mapped into extrinsic representation (context-extrinsic block) but not intrinsic representation (context-intrinsic block). This means that the *Linear* variation falls into the trivial solution. On the contrary, in the *Nonlinear* variation, user information is mapped into extrinsic representation (user-extrinsic block), showing that the extrinsic representation contains both user and context information. Figure 8 shows the performance of the two variations. We can see that *Linear* model performs worse than *Nonlinear* model. It proves that learning intrinsic and extrinsic factors results in a better performance than simply mapping user and context information into two representations, respectively (the trivial solution).

### G.3 COMPARING THE IMPACT OF DIFFERENT CONTRASTIVE LEARNING VARIATIONS

To learn the intrinsic factor, we propose a context-invariant contrastive learning method. However, directly generating intrinsic factor representations through only user information seems to be a more direct way, i.e., $\boldsymbol{o}_{in}^u = f_{ie}^u(\boldsymbol{u})$. We argue that the intrinsic representations learned this way could not guarantee that the representations are intrinsic factors. This is because the information in the learned intrinsic factor representations can vary with different contexts, since these factors have never been modeled w.r.t. the contexts.

In this section, we empirically show that this approach to learning intrinsic factors is inferior to our context-invariant contrastive learning method in producing accurate recommendations. To do so, we design a variation (IEDR$_{sp}$) by splitting the intrinsic-extrinsic factor generation into two functions: $\boldsymbol{o}_{in}^u = f_{in}^u(\boldsymbol{u})$, and $\boldsymbol{o}_{ex}^u = f_{ex}^u(\boldsymbol{u}, \boldsymbol{c})$. Both $f_{in}$ and $f_{ex}$ have the same structure as $f_{ie}$, with the output dimension being a half to ensure the consistency of the factor representation

Table 7: Comparing the performance of IEDR$_{sp}$ with different dropout rates (for *NegGen2*).

|  | Frappe | Yelp |
|---|---|---|
| IEDR$_{sp}$, p=0.1 | 70.68 | 52.03 |
| IEDR$_{sp}$, p=0.5 | 68.25 | 51.49 |
| IEDR$_{sp}$, p=0.1, noDis | 70.56 | 52.02 |
| IEDR$_{sp}$, p=0.1, noCL | 70.31 | 51.62 |
| IEDR$_{sp}$, p=0.1, noCIED | 70.16 | 51.10 |
| IEDR | **74.11** | **53.05** |

Table 8: Comparing the performance of IEDR using different negative context generating methods (for the contrastive learning component).

|  | Frappe | Yelp |
|---|---|---|
| NegGen1 | 73.01 | 52.49 |
| NegGen2 | 71.50 | 51.82 |
| NegGen1&2 | **74.11** | **53.05** |

dimension. The contrastive learning component does not consider context information but uses a standard InfoNCE-based contrastive learning for learning robust user/item representations following (Yao et al., 2021). Table 7 illustrates the results of IEDR$_{sp}$ compared to our model with IEDR$_{sp}$ using different dropout rates ($p = 0.1$ and $p = 0.5$) in the contrastive learning component, and different component combinations (*noDis*, *noCL*, *noCIED*). We can see that our model outperforms the variation in recommendation accuracy. It proves that IEDR$_{sp}$ cannot ensure a successful intrinsic factor learning and hence incur a worse recommendation accuracy. Unlike IEDR, IEDR$_{sp}$ gains better performance with a lower dropout rate. This is because, in IEDR$_{sp}$, the dropout generates views representing the same user instead of different users, which is consistent with the conclusion in (Gao et al., 2021).

### G.4 DIFFERENT NEGATIVE CONTEXT GENERATION METHODS

We propose two negative context generating methods in the contrastive learning component: 1) sample other contexts; 2) use a large dropout rate on the original context. We evaluate the two methods in this section. Table 8 shows the results of our model when using only *NegGens1*, only *NegGens2*, and *NegGen1&2*. We can see that *NegGen1* results in a better performance than using *NegGen2*. This is because *NegGen1* uses true context representations, which are consistent with what may appear in the test samples. Meanwhile, we see that *NegGen1&2* results in the best performance. This is because *NegGen2* provides more unseen (randomly generated) context representations, which strengthens the generalization ability of our model. Next, we evaluate *NegGen2* with different dropout rates in Figure 9. The best performance can be achieved when the dropout rates range from 0.5 to 0.7. This is consistent with our claim in Section 4.2.1. The reason is that a small dropout rate (e.g., 0.1) pushes the generated context representation too close to the original one; hence it cannot be considered a different context. However, a relatively large dropout rate (e.g., 0.9) loses too much information; hence, it is no longer a valid context representation. In addition, for *NegGen1&2* of all the dropout rates, the results consistently outperform those that only use *NegGen2*.

### G.5 EMPIRICAL ANALYSIS OF TIME COMPLEXITY

We summarize the overall time consumption of IEDR and several feature interaction-based baseline models in Table 9. The results are recorded by running the models for one batch (batch size 1024) on the Frappe dataset on a machine with CPU:12th Gen Intel(R) Core(TM) i9-12900K, RAM: 32GB, GPU: NVIDIA GeForce RTX 3090.

We can see that our model's overall time consumption is slightly higher than other baselines. Next, we summarize the time cost of critical procedures in IEDR in Table 10. The first four rows are model forwarding procedures, and the last two rows are model (alternative) optimizing procedures.

Table 10 shows the feature interaction modeling procedure takes most of the time, which is consistent with our analysis above. Both CICL and disentangling forward procedures (row 2-4) do not pose much overhead since they reuse the feature interaction modeling results. Optimization (step1) is the alternative training step that updates the parameters of the models disentangling component ($q_1$ and $q_2$). The alternative optimizing procedure produces little overhead (2.21 ms), which is negligible in the whole procedure.

Table 9: The overall time consumption of different models in one batch training.

| Model | Time (ms) |
| --- | --- |
| DCNv2 | 34.40 |
| AutoInt | 37.53 |
| SIGN | 40.41 |
| IEDR | 44.61 |

Table 10: The time consumption of critical procedures in IEDR in one batch training.

| Procedure | Time (ms) |
| --- | --- |
| Graph Learning (Feature Interaction Modeling) | 14.16 |
| CICL | 8.05 |
| Disentangling (step 1) | 0.16 |
| Disentangling (step 2) | 1.93 |
| Optimization (step 1) | 2.21 |
| Optimization (step 2) | 8.52 |

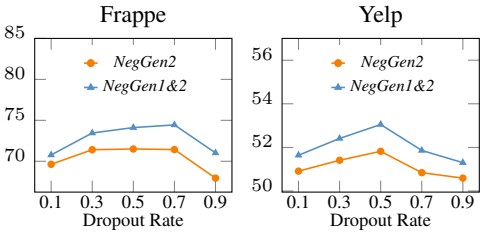

Figure 9: The performance of different dropout rates for the method 2 (*NegGen2*).

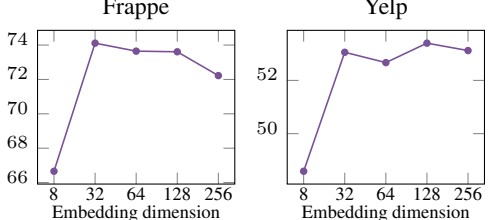

Figure 10: Hyperparameter study: different embedding dimensions $d$.

## G.6 EFFECTIVENESS OF MODEL HYPER PARAMETERS

We evaluate our model with different hyperparameter settings, including embedding dimensions, number of negative samples, and loss weight values. Below, we summaries our observations.

### G.6.1 EMBEDDING DIMENSION

We run our model with different feature embedding dimensions. We show the results of running our model using different embeddings in Figure 10. Choosing the embedding dimension is a trade off between the expression ability and efficiency. From the figure, we can see that larger dimensions result in better prediction accuracy. However, the improvement is not significant when the dimension is larger than 32. A larger dimension may even reduce the performance due to the overfitting problem (e.g., dimension 256 for the Frappe dataset).

### G.6.2 THE NUMBER OF NEGATIVE SAMPLE AND LOSS WEIGHT

The contrastive learning and disentangling components are both contrastive-based methods that require negative sampling. This section evaluates how the number of negative samples influences performance. We also compare the influence of different loss weights of the two components. We run our model with different negative sample numbers and loss weights for the two components, respectively. From Figure 11, we can see that a large loss weight, or a large number of negative samples does not necessarily result in a better performance. There is the best combination of the loss weight and the number of negative samples for both components. A large or small loss weight may make the multi-task training unbalanced, harming the final performance. For the number of negative samples, a small number will make the training insufficient, while a large number may cause an overfitting problem.

### G.7 MORE VISUALIZATIONS OF INTRINSIC AND EXTRINSIC REPRESENTATIONS

This section provides complete intrinsic and extrinsic representation visualizations in three variations: 1) the contrastive learning component is removed (*noCL*); 2) the disentangling component is removed (*noDis*); and 3) the asymmetric disentanglement method (*vCLUB*) is used. Figure 12 compares these results. We include our main observations below:

• The intrinsic and extrinsic factors are perfectly disentangled with our CIED module (*IEDR*).

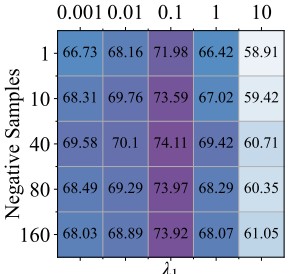 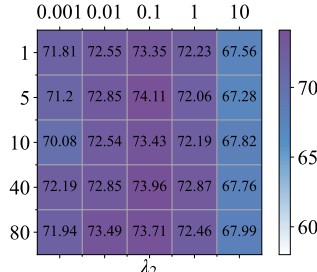

Figure 11: The performance of different numbers of negative samples and the loss weights in the risk minimization function for the contrastive learning component (left) and the disentangling component (right), respectively.

- Without the disentangling component (*noDis*), the intrinsic and extrinsic disentangling procedure may not succeed. This is because there is no restriction on extrinsic representations. Therefore, the extrinsic representations can contain any information, including the information of the intrinsic factor.

- *noCL* has worse disentangling performance than *IEDR*, either. This is because the factors disentangled in *noCL* are implicit. The implicit factors only ensure the disentanglement between the factors of the same data sample, but not between the factors of other data samples. For example, some context information may be stored in the intrinsic representation in data sample 1 but be stored in the extrinsic representation in data sample 2.

- *noCIED* performs worst among all variations, which is reasonable since it does not distinguish the intrinsic and extrinsic representations.

- *vCLUB* performs disentanglement, but is not very stable in some situations. This is consistent with our analysis in Section 4.2.2.

# H ALGORITHM

This section provides the training process of our IEDR model in Algorithm 1. In each epoch, we use the batch stochastic gradient descent method.

---

**Algorithm 1** Batch stochastic gradient descent training of IEDR.

---

1: **Input:** $\mathcal{D} = \{(u_i, v_i, c_i)\}_{i=1:N}$ with the corresponding true label $y_i$ for each data sample.
2: **Hyperparameters:** $B$: batch size; $L$: negative sample number for the context-invariant contrastive learning component; $L_{dis}$: negative sample number for the disentangling component.
3: **Parameters:** $\boldsymbol{\theta}_1^u, \boldsymbol{\theta}_2^u, \boldsymbol{\theta}_1^v, \boldsymbol{\theta}_2^v$: parameters for $q_1^u, q_2^u, q_1^v, q_2^v$, respectively; $\boldsymbol{\omega}$: parameters of IEDR except for $\boldsymbol{\theta}_1^u, \boldsymbol{\theta}_2^u, \boldsymbol{\theta}_1^v, \boldsymbol{\theta}_2^v$.
4: **function** CONTRASTIVELEARNING_USER($\{(\boldsymbol{u}_i, \boldsymbol{c}_i)\}_{i=1:B}$)
5:     **for** $i = 1, ..., B$ **do**
6:         $(\boldsymbol{o}_{in}^u)_{ii} \leftarrow f_{ie}^u(\boldsymbol{u}_i, \boldsymbol{c}_i)$
7:         $ContextGen \leftarrow RandomChoice(NegGen1, NegGen2)$
8:         $c_j \leftarrow ContextGen(c_i)$
9:         $(\boldsymbol{o}_{in}^u)_{ij} \leftarrow f_{ie}^u(\boldsymbol{u}_i, \boldsymbol{c}_j)$               ▷ Generate positive samples.
10:         **for** $\ell = 1, ..., L$ **do**             ▷ Generate negative samples.
11:             $u_{\ell_1} \leftarrow randomChoice(\{u_i\}_{i=1:B}), (\boldsymbol{o}_{in}^u)_{\ell_1 i} = f_{ie}^u(\boldsymbol{u}_{\ell_1}, \boldsymbol{c}_i)$
12:             $u_{\ell_2} \leftarrow randomChoice(\{u_i\}_{i=1:B}), (\boldsymbol{o}_{in}^u)_{\ell_2 j} = f_{ie}^u(\boldsymbol{u}_{\ell_2}, \boldsymbol{c}_j)$
13:         **end for**
14:         $\mathcal{L}_{CICL}(u_i, c_i) \leftarrow$ Equation (4) based on the above positive and negative samples
15:     **end for**
16:     **return** $average(\{\mathcal{L}_{CICL}(u_i, c_i)\}_{i=1:B})$
17: **end function**
18: **function** CONTRASTIVELEARNING_ITEM($\{(\boldsymbol{v}_i, \boldsymbol{c}_i)\}_{i=1:B}$)
19:     Symmetric to CONTRASTIVELEARNING_USER.
20: **end function**
21: **function** DISENTANGLEMENT_USER($\{(\boldsymbol{u}_i, \boldsymbol{c}_i)\}_{i=1:B}$)
22:     **for** $i = 1, ..., B$ **do**
23:         $(\boldsymbol{o}_{in}^u)_{ii}, (\boldsymbol{o}_{ex}^u)_{ii} \leftarrow f_{ie}^u(\boldsymbol{u}_i, \boldsymbol{c}_i)$
24:         $(\boldsymbol{o}_{ex}^u)_{ii}^{pred} \leftarrow q_{\theta_1}((\boldsymbol{o}_{in}^u)_{ii}), (\boldsymbol{o}_{in}^u)_{ii}^{pred} \leftarrow q_{\theta_2}((\boldsymbol{o}_{ex}^u)_{ii})$    ▷ Generate positive samples.
25:         $a_{pos}^{\rightarrow} \leftarrow MSE((\boldsymbol{o}_{ex}^u)_{ii}, (\boldsymbol{o}_{ex}^u)_{ii}^{pred}), a_{pos}^{\leftarrow} \leftarrow MSE((\boldsymbol{o}_{in}^u)_{ii}, (\boldsymbol{o}_{in}^u)_{ii}^{pred})$
26:         $a_{neg}^{\rightarrow} \leftarrow 0, a_{neg}^{\leftarrow} \leftarrow 0$
27:         **for** $j = 1, ..., L_{dis}$ **do**             ▷ Generate negative samples.
28:             $(\boldsymbol{o}_{in}^u)_r, (\boldsymbol{o}_{ex}^u)_r \leftarrow randomChoice\big(\{((\boldsymbol{o}_{in}^u)_{ii}, (\boldsymbol{o}_{ex}^u)_{ii})\}_{i=1:B}\big)$
29:             $(\boldsymbol{o}_{ex}^u)_r^{pred} = q_{\theta_1}((\boldsymbol{o}_{in}^u)_r), (\boldsymbol{o}_{in}^u)_r^{pred} = q_{\theta_2}((\boldsymbol{o}_{ex}^u)_r)$
30:             $a_{neg}^{\rightarrow} \leftarrow a_{neg}^{\rightarrow} + MSE((\boldsymbol{o}_{ex}^u)_{ii}, (\boldsymbol{o}_{ex}^u)_r^{pred})$
31:             $a_{neg}^{\leftarrow} \leftarrow a_{neg}^{\leftarrow} + MSE((\boldsymbol{o}_{in}^u)_{ii}, (\boldsymbol{o}_{in}^u)_r^{pred})$
32:         **end for**
33:         $(\mathcal{L}_{bi\text{-}appr})_i \leftarrow \frac{1}{2}(a_{pos}^{\rightarrow} + a_{pos}^{\leftarrow})$
34:         $(\mathcal{L}_{Dis})_i \leftarrow \frac{1}{2}(\frac{a_{neg}^{\rightarrow} + a_{neg}^{\leftarrow}}{N_{dis}} - (a_{pos}^{\rightarrow} + a_{pos}^{\leftarrow}))$
35:     **end for**
36:     **return** $average(\{(\mathcal{L}_{bi\text{-}appr})_i\}_{i=1:B}), average(\{(\mathcal{L}_{Dis})_i\}_{i=1:B})$
37: **end function**
38: **function** DISENTANGLEMENT_ITEM($\{(\boldsymbol{v}_i, \boldsymbol{c}_i)\}_{i=1:B}$)
39:     Symmetric to DISENTANGLEMENT_USER.
40: **end function**
41:

---

---

**Algorithm 1** Batch stochastic gradient descent training of IEDR (continued).

---

42: $shuffle(\{(u_i, v_i, c_i)\}_{i=1:N})$
43: **for** *each batch* $\{(u_i, v_i, c_i)\}_{i=1:B}$ **do**
44:     **for** $i = 1, ..., B$ **do**           ▷ Line 45-47 are the recommendation prediction module.
45:         $\boldsymbol{u}_i \leftarrow f_u(u_i), \boldsymbol{v}_i \leftarrow f_v(v_i), \boldsymbol{c}_i \leftarrow f_c(c_i)$
46:         $(\boldsymbol{o}_{in}^u)_{ii}, (\boldsymbol{o}_{ex}^u)_{ii} \leftarrow f_{ie}^u(\boldsymbol{u}_i, \boldsymbol{c}_i), (\boldsymbol{o}_{in}^v)_{ii}, (\boldsymbol{o}_{ex}^v)_{ii} \leftarrow f_{ie}^v(\boldsymbol{v}_i, \boldsymbol{c}_i)$
47:         $y_i' \leftarrow f_{pred}((\boldsymbol{o}_{in}^u)_{ii}, (\boldsymbol{o}_{ex}^u)_{ii}, (\boldsymbol{o}_{in}^v)_{ii}, (\boldsymbol{o}_{ex}^v)_{ii})$
48:         $(\mathcal{L}_{RP})_i \leftarrow CrossEntropy(y_i', y_i)$
49:     **end for**
50:     $\mathcal{L}_{RP} \leftarrow average(\{\mathcal{L}_{RP})_i\}_{i=1:B}$
51:     $\mathcal{L}_{CICL}^u \leftarrow \text{CONTRASTIVELEARNING\_USER}(\{(\boldsymbol{u}_i, \boldsymbol{c}_i)\}_{i=1:B})$
52:     $\mathcal{L}_{CICL}^v \leftarrow \text{CONTRASTIVELEARNING\_ITEM}(\{(\boldsymbol{v}_i, \boldsymbol{c}_i)\}_{i=1:B})$
53:     $\mathcal{L}_{bi\text{-}appr}^u, \mathcal{L}_{Dis}^u \leftarrow \text{DISENTANGLEMENT\_USER}(\{(\boldsymbol{u}_i, \boldsymbol{c}_i)\}_{i=1:B})$
54:     $\mathcal{L}_{bi\text{-}appr}^v, \mathcal{L}_{Dis}^v \leftarrow \text{DISENTANGLEMENT\_ITEM}(\{(\boldsymbol{v}_i, \boldsymbol{c}_i)\}_{i=1:B})$
55:     Freeze $\boldsymbol{\omega}$, update $\boldsymbol{\theta}_1^u, \boldsymbol{\theta}_2^u, \boldsymbol{\theta}_1^v, \boldsymbol{\theta}_2^v$ through minimizing $\mathcal{R}(\boldsymbol{\theta}_1^u, \boldsymbol{\theta}_2^u, \boldsymbol{\theta}_1^v, \boldsymbol{\theta}_2^v)$     ▷ Step 1
56:     Freeze $\boldsymbol{\theta}_1^u, \boldsymbol{\theta}_2^u, \boldsymbol{\theta}_1^v, \boldsymbol{\theta}_2^v$, update $\boldsymbol{\omega}$ through minimizing $\mathcal{R}(\boldsymbol{\omega})$     ▷ Step 2
57: **end for**

---

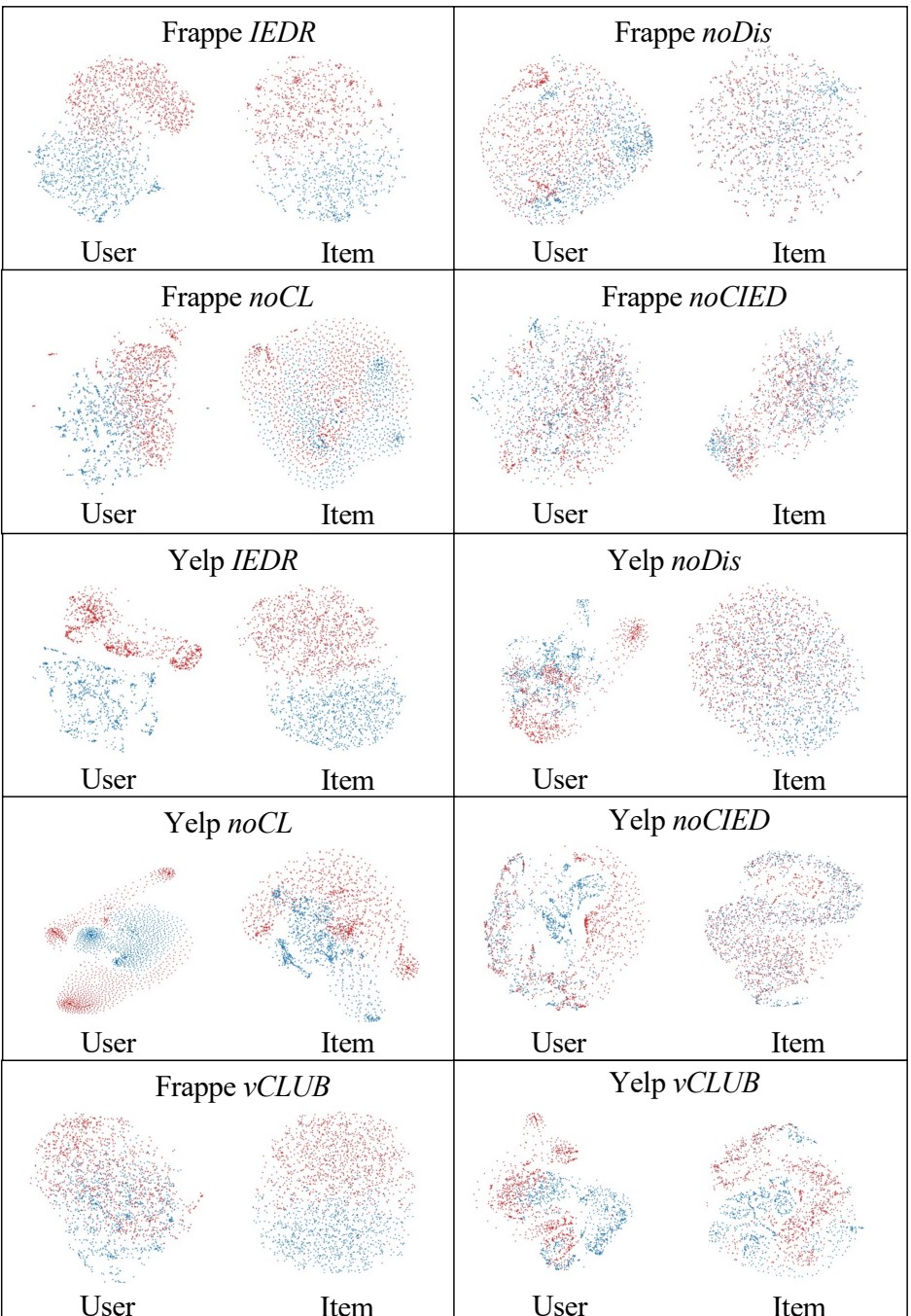

Figure 12: The complete intrinsic-extrinsic disentanglement visualizations in t-SNE. The blue dots are intrinsic representations, and the red dots are extrinsic representations.

