# OpenReview forum: "IEDR: A Context-aware Intrinsic and Extrinsic Disentangled Recommender System"
_ICLR.cc/2023/Conference — Submitted to ICLR 2023_

### Official Review · Reviewer_ywxS · 2022-10-24

**Confidence:** 4
**Correctness:** 3
**Technical Novelty And Significance:** 3
**Empirical Novelty And Significance:** 3
**Recommendation:** 6

**Clarity, Quality, Novelty And Reproducibility:**

Most of the paper is understandable and the paper quality is acceptable. The topic is novel and important. However, the authors do not provide codes and it may lead to difficulties of reproducibility.

**Strength And Weaknesses:**

Strength:
S1. Interesting problem and novel idea in recsys.
S2. Addressing the issue of factors learning in complex contexts and disentangling the factors into intrinsic and extrinsic parts for better recommendation.
S3. Providing some theorem analysis about the optimization problem.

Weaknesses:
W1. Some claims are not clear. For example, in 4.2.1, why choosing the negative pairs from the same context with different users? Are there any assumptions?
W2. The proposed method seems to be complicated. Should give a specific analysis of the time complexity.
W3. Please provide more analysis and experiments on other specific context except for the time context in the section 5.1.2.
W4. There is no comparison with other methods in the section 5.3.2.
W5. The results on Movies and CDs datasets are missing in Table 1.


**Summary Of The Paper:**

The authors discussed intrinsic and extrinsic factors that jointly affect users’ decisions in items selection. This problem is interesting and important. This paper provides a new factor learning method for more contexts that existing studies have not mainly considered.
The contributions are:
C1. Proposing a context-agnostic intrinsic and extrinsic factors learning framework that can tackle different contexts.
C2. Exploiting a context-invariant contrastive learning and a mutual information minimization-based disentangling to disentangle the factors, and providing an explainable results about the different factors.
C3. Providing a theorem analysis about the proposed contrastive loss.
C4. Experiments show the effectiveness of the proposed framework.


**Summary Of The Review:**

This paper tries to solve a novel problem in recommender systems. The motivation and method design are clearly presented. However, one main concern is the flexibility of the method since there are many tasks and hyper-parameters need to be optimized, the tuning process might be very complicated.

---

> ### Author Response · Authors · 2022-11-17
> **Response to Reviewer ywxS**
>
> Thanks for the reviewer's positive and constructive feedback.
>
> ---
> > Some claims are not clear. For example, in 4.2.1, why choosing the negative pairs from the same context with different users? Are there any assumptions?
>
> We choose the negative pairs from the same context with different users based on two reasons:
> 1. Users should have personalized intrinsic factors. The negative pairs emphasize this fact and try to learn personalized intrinsic representations even under the same context.
> 2. From the training perspective, assigning the same context can improve the training efficiency, since the model can focus on differentiating users' personalized intrinsic factors without being influenced by the change of the contexts. Meanwhile, removing the constraint of equal context will dramatically expand the sampling space, and most of the combinations rarely contribute to training (i.e., if two intrinsic factors are generated from different contexts and different users, they will be easily distinguished).
>
> ---
> > The proposed method seems to be complicated. Should give a specific analysis of the time complexity.
>
> Similar questions were asked by Reviewer vJvU. Due to the response space limitation, please see our detailed analysis to this reviewer.
>
> ---
> > Please provide more analysis and experiments on other specific context except for the time context in the section 5.1.2.
>
> Sequential recommendation is the most representative domain to consider intrinsic and extrinsic factors (i.e., long- and short-term interests). Therefore, our evaluation of this domain (Table 2) is significant and convincing to prove our claim.
>
> We value the reviewer's suggestion. Since such experiments are time consuming and the response time is limited, we will consider adding experiments after the response window.
>
> ---
> > There is no comparison with other methods in the section 5.3.2.
>
> Since no baseline can perform intrinsic and extrinsic factor learning on the Frappe dataset, we cannot perform a fair comparison in this section (DGCF can learn implicit factors, but cannot generate item rankings for each factor).
>
> To further illustrate whether the rankings from intrinsic factors and extrinsic factors are reasonable, we show the apps with the top 10 intrinsic and extrinsic scores of User1 on weekdays below.
>
> ||Intrinsic|Extrinsic|
> |--- |---|---|
> |1|Photography|Tool|
> |2|Sports| Communication|
> |3|Health&Fitness|Media&Video|
> |4|Tools|Personalization|
> |5|Health & Fitness|Communication|
> |6|Personalization|Casual|
> |7|Personalization|Music&Audio|
> |8|Communication|News&Magazines|
> |9|Personalization|Communication|
> |10|Health & Fitness|Travel&Local|
>
> It shows that the Tool (Google Search) and Communication (Gmail) rank high in extrinsic scores, which are often used while working. However, the intrinsic scores show more about users' real interests, e.g., Photography (Magical Video Editor) and Sports (Football Scores Live). We included the results in Section 5.3.2.
>
> ---
> > The results on Movies and CDs datasets are missing in Table 1.
>
> We have evaluated the three best-performed baselines in Table 1 on the Movies and CDs datasets. The results are in the table below.
> These baselines do not disentangle intrinsic and extrinsic factors. Therefore, they perform worse than our models like the Frappe and Yelp datasets.
>
> |   Model    | Movies (AUC)  | Movies (N@10)    |CDs (AUC) | CDs (N@10) |
> | ------- | -------| ------- | -------- | -------|
> | AutoInt | 77.78 | 22.27 | 77.65 | 18.25 |
> | DeepFM  | 78.50 | 23.13 | 78.26 | 19.18 |
> | DCNv2   | 78.63 | 23.04 | 78.30 | 19.33 |
> | SIGN    | 78.82 | 23.58 | 78.95 | 19.97 |
> | IEDR    | 80.14 | 26.68 | 80.34 | 20.95 |
>
> ---
> > the authors do not provide codes and it may lead to difficulties of reproducibility.
>
> Apologize for not being able to release our code at the time of our paper submission. This is because at the time of submission, we did not pass the internal code review for data desensitization within our organization, and hence, we were not allowed to release the code. Fortunately, now, we have passed it and our code can be found in the anonymous repository at: https://anonymous.4open.science/r/IEDR-20E5/
>
> ---
> > One main concern is the flexibility of the method since there are many tasks and hyper-parameters need to be optimized, the tuning process might be very complicated.
>
> Our hyper-parameter studies in Appendix F.4 and F.5 show that most of the hyper-parameter settings can be fixed for all the datasets (e.g., dropout rates and numbers of negative samples) to yield a good result.
> The hyper-parameters that significantly influence the training performance are the weights ($\lambda_1$ and $\lambda_2$), which balance the multi-task training. Tuning these weights is required in most multi-task training models. Our experimental results show that the best weight setting falls into a small range (i.e., 0.01~0.1), which eases the hyperparameter tuning.

---

### Official Review · Reviewer_N1Ar · 2022-10-24

**Confidence:** 4
**Correctness:** 3
**Technical Novelty And Significance:** 2
**Empirical Novelty And Significance:** 3
**Recommendation:** 6

**Clarity, Quality, Novelty And Reproducibility:**

The research problem is well-motivated and extensive experiments prove the effectiveness of the proposed method. The paper is innovative, but not sufficient. The authors do not give a detailed description of their experiment setting for reproducibility.

**Strength And Weaknesses:**

Strength:
1.	The paper proposes a good perspective to learn disentangled representations using contrastive learning.
2.	It is interesting to use psychological theory to divide users' purchasing decisions into intrinsic and extrinsic reasons
3.	The rich experiment results prove the effectiveness of the method.

Weakness:
1.	The proposed method learns two types of disentangled representations. I’m curious about how authors can distinguish whether they are intrinsic or extrinsic factors.
2.	Some technics are based on previous work. Despite some improvement, it is still not innovative enough, e.g., existing SIGN, vCLUB, using contrastive learning to learn disentangled representations.
3.	About the experiment part, when comparing the sequential recommending methods, there is no detailed description of the settings of the proposed method, like how they process.



**Summary Of The Paper:**

The paper proposes a generic model to learn intrinsic and extrinsic factors from various contexts using a contrastive learning component, and a disentangling component. Experimental results on real-world datasets demonstrate the effectiveness of the method. Additionally, theoretical proofs are provided for the refined vCLUB.

**Summary Of The Review:**

The research problem is well-motivated. The authors give completed theoretical proofs and conduct rich experiments. But the technic is lack innovation.

---

> ### Author Response · Authors · 2022-11-17
> **Response to Reviewer N1Ar**
>
> We thank the reviewer's positive and useful feedback.
>
> ---
> > I'm curious about how authors can distinguish whether they are intrinsic or extrinsic factors.
>
> The key to distinguishing the intrinsic and extrinsic factors is our proposed context-invariant contrastive learning component (Section 4.2.1). The component helps to capture users'/items' information that is not influenced by different contexts. The captured information is context-invariant (intrinsic factors) and stored in intrinsic factor representations. Meanwhile, other useful information that is influenced by different contexts is captured and stored in extrinsic factor representations.
>
>
> ---
> > Some technics are based on previous work. Despite some improvement, it is still not innovative enough.
>
> The main contribution is that this paper is the first work to propose a general framework to capture intrinsic and extrinsic factors from any number and any type of context. To achieve this, we propose a novel contrastive learning approach (Section 4.2.1) to capture intrinsic factors. Technically, the context-invariant contrastive learning approach provides insight into how to leverage the contrastive learning framework to disentangle true signals (intrinsic factors) from noisy information (use/item attributes) based on critical conditions (contexts). Connections to information theory are provided to further understand the approach (Theorem 1). Meanwhile, we propose a simple yet effective improvement of vCLUB as our disentangling component, which has been theoretically (Appendix D) and empirically (Section 5.2.2) proven a better disentanglement method than the state-of-art vCLUB method.
>
> ---
> > When comparing the sequential recommending methods, there is no detailed description of the settings of the proposed method, like how they process.
>
>
> The Amazon CDs and Movies are datasets for sequential recommendations. We directly use the datasets to train the sequential recommendation baselines.
>
> In IEDR, we use the **same** model structure as that for the Frappe and Yelp datasets, but modify the data input to fit our model. More specifically, we do not directly learn behavior sequences, but consider each behavior as a data sample with time context information.
> That is, we consider the bucketed timestamp of each user behavior as a time context (we consider one month as a categorized time context). Therefore, behaviors in the same time interval have the same time context, indicating that these behaviors share some similar short-term (extrinsic) interests (e.g., item popularity).
>
> We included this in Appendix F. Meanwhile, we have published the code anonymously at: https://anonymous.4open.science/r/IEDR-20E5/.

---

### Official Review · Reviewer_TrQ3 · 2022-10-25

**Confidence:** 3
**Clarity, Quality, Novelty And Reproducibility:** The clarity and quality are good. The…
**Correctness:** 3
**Technical Novelty And Significance:** 3
**Empirical Novelty And Significance:** 3
**Recommendation:** 6

**Strength And Weaknesses:**

S1. Overall, the paper is well-organized and well-written. The main idea is clearly illustrated and the figures are quite helpful for understanding the proposed IEDR method.

S2. They conduct extensive experiments on two public datasets, including performance comparison, ablation study, visualization, and case study. The main comparison results are promising and the other results meet their expectations.

---
W1. The main concern is about the motivation of the proposed method. The paper claims that intrinsic factors are user-related (or item-related) factors that don't change with context, while extrinsic factors are contextual features that don't change with user or item.

Then it is straightforward to just use user-related and item-related features as intrinsic factors, and use contextual features as extrinsic factors. It doesn't bring any benefit to getting the intrinsic factors by extracting context-free factors from the input of both contextual features and user (item) features. Thus the method seems to be superfluous as it is trying to seek an inferior answer with the best answer in mind.

I have noticed the clarification of preventing the trivial solution in Appendix C. But I don't think it is a trivial solution to map user features to intrinsic factors and to map contextual features to extrinsic factors, because they are just the right answer according to both the motivation and the proposed method.

W2.  Although the paper conducted experiments on two public datasets, the used datasets are both very small datasets with less than one million samples, which hurts the convincing of the comparison results. It is strongly suggested to add results on benchmark CTR prediction datasets like Avazu or Taobao.

**Summary Of The Paper:**

This paper proposes an intrinsic-extrinsic disentangled model named IEDR for recommendation. The idea is that user-related (or item-related) features are intrinsic factors that don't change with time, while contextual features are extrinsic factors that change with time.

Thus they aim to get the intrinsic factors by maximizing the agreement between the encoded representation of the same user under different contexts, and minimizing the agreement between the encoded representation of the same context with different users. This is done by optimizing an InfoNCE loss in the contrastive learning module.

Besides, they also try to disentangle extrinsic factors by minimizing the mutual information between the extrinsic factors and the above-generated intrinsic factors.

Experiments are conducted on two public datasets to conclude the superior performance of IEDR in contrast to other CTR prediction models.

**Summary Of The Review:**

Due to the weakness in motivation and evaluation, I would vote for rejection.

---
AFTER THE AUTHOR RESPONSE:

The author's response does help to solve my confusion about the motivation of the proposed method.

I'm still a bit worried about the convincing of the experimental results.  The authors explain that Taobao and Avazu datasets are not suitable for the scenario because they only span less than two weeks. But there are some other available benchmark datasets like Movielens-20M with timestamp features that span decades. My suggestion is that at least the main comparison result should also be provided for the two Amazon datasets in addition to Frappe and Yelp.

Due to the above points, I would like to raise my score to marginally above the acceptance threshold.

---

> ### Author Response · Authors · 2022-11-17
> **Response to Reviewer TrQ3**
>
> We thank the reviewer's valuable comments.
>
> ---
> > Then it is straightforward to just use user-related and item-related features as intrinsic factors, and use contextual features as extrinsic factors. It doesn't bring any benefit to getting the intrinsic factors by extracting context-free factors from the input of both contextual features and user (item) features.
>
> To answer the question, we need to carefully differentiate (intrinsic factors, extrinsic factors) from (user/item information, contextual information).
>
> It has been shown that disentangling intrinsic and extrinsic factors enables a better understanding of users' decisions [1]. However, careful design is required to successfully disentangle and capture the two kinds of factors.
>
> Briefly speaking, the intrinsic/extrinsic learning in our model is motivation-level learning that captures only the part of user/item information (motivation) that is **insensitive to contexts** as the intrinsic factor. The other part of the user/item information that is **sensitive to contexts** will be combined together with contextual information as the extrinsic factor. However, the suggested method is feature-level learning that maps **all** the user/item-related information (entangles both context-insensitive and context-sensitive information) into one representation (the "intrinsic factor" in the suggested method), and maps the contextual information into another representation (the "extrinsic factor" in the suggested method). These differences are demonstrated in Appendix C (Eq. 14 vs. Eq. 15).
>
> In detail, from the *intrinsic* factor's perspective, the suggested method that "just use user-related and item-related features as intrinsic factors, and use contextual features as extrinsic factors" still entangles the user/item information (motivations) that are sensitive or insensitive to the contextual information. This is against our definitions of intrinsic/extrinsic factors based on psychological research. Instead, our IEDR approach successfully disentangles the two motivations to learn intrinsic and extrinsic factors.
>
> From the *extrinsic* factor perspective, the straightforward way will end up with the same extrinsic factor representation among all the users and items. It means all the users respond the same to the same context. However, it is intuitive that different users may respond differently to the same context (e.g., user A will take an Uber to work on a rainy day, while user B will work from home on a rainy day). Our IEDR approach captures such differences by taking the user/item information (sensitive to context) and contextual information to learn personalized extrinsic factors.
>
> Therefore, the essential differences we discussed above result in the superior performance of the IEDR model over the trivial situations in Appendix F.2.
>
> In Appendix F.3, we also evaluate a model variation that is similar to the suggested method: directly use user/item-related features as intrinsic factors, and use user/item-related features and contextual features together as extrinsic factors. This variation still entangles context-sensitive/insensitive information in the intrinsic representations and results in inferior performance.
>
> [1] Roland B'enabou and Jean Tirole. Intrinsic and extrinsic motivation. The Review of Economic Studies, pp. 489–520, 2003.
>
> ---
> > The used datasets are both very small datasets with less than one million samples, which hurts the convincing of the comparison results.
>
> We evaluated our model on four public datasets, as summarized in the results in Table 4 in the Appendix (page 17). Apart from Frappe and Yelp, we also included representative experimental results of the Amazon Movies dataset (with millions of samples in total) and Amazon CDs dataset, as demonstrated in Table 2 (page 7).
>
> We did not evaluate our model on Taobao and Avazu datasets because they are not suitable for our scenario. Although they have several item-related or/and user-related features, the contextual features are not available. One exception is the timestamp feature as context. However, the timestamp feature covers only 11 days for Avazu and 8 days for Taobao, which cannot provide the necessary information for disentanglement learning.
>
> In addition, large CTR datasets such as Taobao and Avazu contain large discrepancies (noises) between users’ responses and their true interests, which may cause an inaccurate evaluation of representation learning. Amazon and Yelp datasets are review/comment-based datasets. The quality of these datasets can be guaranteed even when the sample size is not very huge. On the contrary. Therefore, our used datasets are more suitable to evaluate our key motivation: learning better intrinsic/extrinsic factor representations.

---

### Official Review · Reviewer_vJvU · 2022-10-29

**Confidence:** 3
**Correctness:** 3
**Technical Novelty And Significance:** 3
**Empirical Novelty And Significance:** 3
**Recommendation:** 6

**Clarity, Quality, Novelty And Reproducibility:**

This paper is well written and clearly presented.

The code should be made public to validate *reproducibility* owing to the fact that implementation details are missing in the text.

*Quality* and *novelty* is convincing (see Strength and Weaknesses).

**Strength And Weaknesses:**

This paper is well written and clearly presented.

The idea of splitting up context-invariant intrinsic factors and context-based extrinsic factors are interesting and inspiring.

The mutual information minimization-based disentangling component looks over-designed. This component directly introduces the alternative optimization procedure. As such, a natural question is not addressed: does IEDR suffer from more computational overhead? This issue should be carefully discussed and also empirically validated in the experiment section.

The implementation of IEDR in Section 5.1.2 is not sufficiently clear. Is the user behavior sequence not used as a context for IEDR? If so, the experimental results of this section is surprising in some sense.

**Summary Of The Paper:**

This paper proposes a method (IEDR) for recommendation systems which explicitly models the intrinsic representations and the extrinsic representations. Intrinsic representations and extrinsic representations are invariant and variant to context features, respectively; a proposed contractive learning component and a mutual information minimization-based disentangling component are collaboratively utilized. Experiments on public datasets demonstrate IEDR’s effectiveness over the state-of-the-art methods.

**Summary Of The Review:**

See “Strength And Weaknesses” and “Clarity, Quality, Novelty And Reproducibility”.

---

> ### Author Response · Authors · 2022-11-17
> **Response to Reviewer vJvU**
>
> We thank the reviewer's positive and insightful feedback.
>
> ---
> > Does IEDR suffer from more computational overhead?
>
> We analyzed the time complexity of IEDR qualitatively and quantitatively.
>
> Briefly speaking, the time complexity of the whole model is comparable to feature interaction-based recommender systems (e.g., AutoInt, SIGN). The overhead of the alternative optimizing procedure for the disentanglement component is marginal in the whole optimizing procedure. The detailed analysis is shown below.
>
> *Qualitatively analysis:*
>
> The most time-consuming step of IEDR is the feature interaction learning to get user, item, and context representations, which needs to model on every pair of feature interactions. This procedure has also been done on other feature interaction-based models. Therefore, the time complexity of the proposed module is comparable with those methods.
>
> Our model takes additional computations on the contrastive learning component (CICL) and the disentangling component:
>
> For the CICL component, we do not need to perform the feature interaction modeling again, but reuse the generated user/item/context representations, which saves the majority of the overhead. We only need to perform $f_{ie}$ (a one-hidden layer MLP with 128 hidden units) for factor generating, hence the overhead is marginal.
> For the disentangling component, we also reuse the generated user/item/context representations.
>
> The disentangling component introduces the two-step learning policy to train our model.
> Regarding to the reviewer's main concern, the first step in the two-step learning actually takes very little overhead. This is because this step only tries to optimize the parameters of functions $q_1$ and $q_2$ (Eq.2), which are two MLPs with one hidden layer. For each data sample, we only run $q_1$ and $q_2$ for one time.
>
> In summary, all of the computations above can reuse the feature interaction modeling results, the small imposed overhead is acceptable considering the effectiveness of our model in capturing accurate intrinsic and extrinsic factors.
>
> *Quantitatively evaluation:*
>
> We summarize the overall time consumption of IEDR and several feature interaction-based baseline models in the table below. The results are recorded by running the models for one batch (batch size 1024) on the Frappe dataset on a machine with CPU:12th Gen Intel(R) Core(TM) i9-12900K, RAM: 32GB, GPU: NVIDIA GeForce RTX 3090.
>
> | Model      | Time (ms) |
> | ----------- | ----------- |
> | DCNv2 | 34.40  |
> | AutoInt | 37.53 |
> |  SIGN  | 40.41   |
> |  IEDR | 44.61  |
>
> We can see that our model's overall time consumption is slightly higher than other baselines. Next, we summarize the time cost of critical procedures in IEDR (other procedures include embedding lookup and the merge of the factors). The first four rows are forwarding procedures, and the last two rows are optimizing procedures.
>
> | Procedure      | Time (ms) |
> | ----------- | ----------- |
> | Graph Learning (Feature Interaction Modeling) | 14.16 |
> |  CICL  | 8.05   |
> |  Disentangling (step 1)  | 0.16 |
> |  Disentangling (step 2)  | 1.93  |
> |  Optimization (step 1)     | 2.21  |
> |  Optimization (step 2)     | 8.52  |
>
> We can see that the feature interaction modeling procedure takes most of the time.
> Both CICL and disentangling forward procedures (rows 2-4) do not pose much overhead since they reuse the feature interaction modeling results.
> Optimization (step1) is the alternative training step that updates the parameters of the model's disentangling component ($q_1$ and $q_2$). The alternative optimizing procedure produces little overhead (2.21 ms), which is negligible in the whole procedure.
>
> We included the above analyses in Appendix E and Appendix G.5.
>
> ---
> > The implementation of IEDR in Section 5.1.2 is not sufficiently clear. Is the user behavior sequence not used as a context for IEDR?
>
> In IEDR, we use the **same** model structure as that for the Frappe and Yelp datasets, but modify the data input to fit our model. More specifically, we do not directly learn behavior sequences, but consider each behavior as a data sample with time context information.
> That is, we consider the bucketed timestamp of each user behavior as a time context (we consider one month as a categorized time context). Therefore, behaviors in the same time interval have the same time context, indicating that these behaviors share some similar short-term (extrinsic) interests (e.g., item popularity).
> We included this in Appendix F.
>
> ---
> > The code should be made public to validate reproducibility
>
> Apologize for not being able to release our code at the time of our paper submission. This is because at the time of submission, we did not pass the internal code review for data desensitization within our organization, and hence, we were not allowed to release the code. Fortunately, now, we have passed it and our code can be found in the anonymous repository at: https://anonymous.4open.science/r/IEDR-20E5/

---

### Decision · Program_Chairs · 2023-01-20

**Decision:**

Reject

**Justification For Why Not Higher Score:**

Overall the reviews on this paper are relatively lukewarm for the lack of a better word. The strengths and weaknesses of the paper are pretty unanimously agreed among the reviewers. I have also read the paper myself in the past week. This is by no means a bad paper. But I think the execution on separating the intrinsic and extrinsic factors, as well as the experimental setup (the first two weaknesses above), are the main reasons that I did not give a higher score.

**Justification For Why Not Lower Score:**

N/A

**Metareview: Summary, Strengths And Weaknesses:**

Summary: This paper proposes IEDR which learns both intrinsic and extrinsic factors for contextual recommender systems (e.g., context can be weather, devices...). A user/item intrinsic factor is context-invariant for a given user/item, while an extrinsic factor will differ based on different contexts for a given user/item. In order to enforce these model assumptions, the authors propose a contrastive learning objective for intrinsic factors: given a user, maximize the agreement between different contexts while minimize the agreement between different users with the same context. In addition, the authors add a mutual information minimization objective to encourage disentanglement between the intrinsic and extrinsic factors. Experimental results show that IEDR performs favorably over other similarly-structured factorization (some of them only implicitly factorize) models.

Strength: The reviewers all agree that the separation between intrinsic and extrinsic factors makes sense as a way to capture different aspect of user preference/item attributes. The connection to psychological research is an interesting perspective. The paper is mostly well written.

Weakness: Some of the minor weaknesses (source code, model training time) have been addressed during the author response period. A few unresolved concerns from the reviewers include (I have also read the paper myself in the last week due to the paper being borderline so some of these concerns are also shared/expanded by me below):
* The separation between intrinsic and extrinsic factors, though conceptually straightforward, remains questionable in execution (reviewer TrQ3, N1Ar and ywxS). The definition of intrinsic factors is mathematically clear and it is understandable how a contrastive loss would be useful here (even though there are still some debatable parts about how exactly the sampling is carried out, mostly from reviewer ywxS; I also personally found Eq 1 unclear: it should either be a function of $c_j$ since $c_j$ appears on the numerator or (more likely) take expectation over $c_j$, in which case this expectation is approximated by sampling a different context. Furthermore, Eq 1 is not exactly InfoNCE given that the numerator is not included in the normalizing factor on the denominator). On the other hand, the definition of the extrinsic factors is rather ambiguous. From definition 1, every function $f$ that does not produce an intrinsic factor (i.e., there exist $c_i$ and $c_j$ such that $f(u, c_i) \neq f(u, c_j)$ would be considered as an extrinsic factor, which is too broad to be useful. A natural question arises: why would you choose such a mutual information minimization objective and not something else? I can see these additional disentanglement help as a form of regularization/prior, but that is not because of the separation between intrinsic and extrinsic factors. It almost feels like the authors are too fixated on the narrative of "intrinsic and extrinsic" even though it doesn't exactly fit the paper perfectly. I would feel more comfortable with the extrinsic factors if the authors can also define it from an invariant perspective similar to intrinsic factors. The current definition of "B is everything that is not A" is not very helpful with defining and justifying the learning objective.
* Concerns about the experimental setup. Reviewer TrQ3 points out the main results are only reported on two very small datasets (Frappe and Yelp) and even the relative larger datasets (Amazon Movies and CDs) are still comparably small among the overall recommender systems literature. This is something I would also agree with -- in recommender systems, it can often be observed that a method which works well on smaller datasets would not necessarily excel on realistically-sized datasets.
* Reviewer N1Ar concerns that the technics used in the paper are most from existing work which lacks innovation. I think this is a fair observation but I don't think it as a main weakness (especially comparing with the two above).